# Equitable Federated Learning with Activation Clustering

## Abstract

Federated learning is a prominent distributed learning paradigm that incorporates collaboration among diverse clients, promotes data locality, and thus ensures privacy. These clients have their own technological, cultural, and other biases in the process of data generation. However, the present standard often ignores this bias/heterogeneity, perpetuating bias against certain groups rather than mitigating it. In response to this concern, we propose an equitable clustering-based framework where the clients are categorized/clustered based on how similar they are to each other. We propose a unique way to construct the similarity matrix that uses activation vectors. Furthermore, we propose a client weighing mechanism to ensure that each cluster receives equal importance and establish $\mathcal{O}\left(\frac{1}{\sqrt{K}}\right)$ rate of convergence to reach an $\epsilon$-stationary solution. We assess the effectiveness of our proposed strategy against common baselines, demonstrating its efficacy in terms of reducing the bias existing amongst various client clusters and consequently ameliorating algorithmic bias against specific groups.

## 1 Introduction

With the advent of distributed learning paradigms, Federated Learning (FL) emerged as a promising mechanism for collaborative learning among diverse clients. In FL, the diversity of the clients gives rise to both system and statistical heterogeneity McMahan et al. (2017); Li et al. (2020). The system heterogeneity is attributable to the different computational capabilities of the participating clients, uplink and downlink communication channel bandwidths, and faults. Statistical heterogeneity exists due to the variability in the data distribution coming from the different clients. In our work, our primary aim is to tackle the statistical heterogeneity in the data distribution arising due to the technological, cultural, and other biases in the data generation process native to each client.

In a distributed learning setting, algorithms use the participating clients' local updates and perform global weighted aggregation. The goal is to create a global model that performs equally well on all clients. This leads to a notion of group fairness Dwork et al. (2012) that incentivizes the participation of diverse groups of clients Zhan et al. (2021), thus mitigating the bias in performance during the learning process. The end goal of such algorithms is to ensure that the generated model is not biased towards any particular group of clients and performs well for all of them. This is achieved through various techniques, such as FL, where the model is trained on the data from all participating clients without compromising the privacy of individual clients. Ultimately, the success of distributed learning algorithms depends on their ability to balance the needs and preferences of all participating clients to generate a global model that is accurate, efficient, and fair. A celebrated FL method, FedAvg McMahan et al. (2017), showcases success in the case of IID distribution of data among the clients. However, its performance in the case of non-IID data distribution is the subject of interest. Several studies such as Li et al. (2020); Karimireddy et al. (2020) have been conducted and provide better convergence of FL on non-IID scenarios. Although these algorithms provide better results and tackle the problem of data heterogeneity, they do not account for the inherent structure among the participating clients.

This paper presents a novel approach toward dealing with the non-IID scenario in FL. It proposes a clustering paradigm based on *activation vectors (See Definition 1)* that promote group fairness. In the context of FL, the idea of clients belonging to a cluster is not a strict assumption. Instead, it is already present in the current

paradigm, for instance, when clients from different countries participate. In this broader context, the clients representing distinct clusters, such as different countries, can benefit from the shared knowledge in FL. The primary goal of the proposed algorithm is to assign equal weightage to each participating cluster, where each cluster contains a varying number of clients, for global model aggregation. This approach promotes an equitable and fair treatment of clients, utilizing the underlying similarities among the participating clusters. Therefore, it can be considered as a server-side debiasing method, ensuring all clients benefit from the shared knowledge.

## 1.1 Contribution

Our contributions are summarized as follows:

- Our proposed solution is a novel clustering framework that employs activation vectors as the primary mechanism to group clients based on their similarities. This approach addresses the challenges that arise when clients from diverse backgrounds participate in the clustering process. By using activation vectors, our framework can effectively capture each client's unique characteristics and preferences while also identifying commonalities that allow for meaningful clustering.

- Our proposed approach involves leveraging this side information to enhance the model aggregation process. This novel scheme can be seamlessly integrated with any existing client scheduler, allowing for improved performance and more efficient utilization of resources.

- We present the convergence analysis for our algorithm, termed `Equitable-FL`, and show that it enjoys a convergence rate of $\mathcal{O}\left(\frac{1}{\sqrt{K}}\right)$ to reach an $\epsilon$-stationary solution under mild assumptions.

- We have conducted thorough experiments to compare our framework against baseline algorithms using popular vision datasets like MNIST, CIFAR-10, CIFAR-100, and FEMNIST. Our findings demonstrate that our algorithm not only delivers high accuracy, but it also minimizes client disagreements, thus mitigating the algorithmic bias against certain groups.

## 1.2 Organization

The rest of the paper is organized as follows. Section 2 discusses the related work. Section 3 introduces the relevant concepts and preliminaries. Section 4 introduces our proposed approach and discusses its convergence properties. Section 5 corroborates our method through extensive experiments while concluding remarks are stated in Section 6.

## 2 Related Work

**Centralized and Consensus-based Methods.** Fairness has long been a concern in machine learning. Traditional centralized machine learning approaches use pre-processing and post- processing techniques to ensure fairness since the central server typically has access to the data Grgić-Hlača et al. (2018); Zhang et al. (2018); Lohia et al. (2019); Kim et al. (2019); Mehrabi et al. (2021). However, these methods do not suit paradigms like FL, which prioritize data locality and privacy. Achieving fairness in FL remains a critical research area. This paper focuses on traditional FL settings where collaborative model training happens across multiple clients while preserving the data privacy. Researchers have proposed extensions to FedAvg McMahan et al. (2017), including model-level regularization Li et al. (2020); Durmus et al. (2021), feature alignment between local and global models Li et al. (2021), alignment of local and a global meta-model via gradient correction Acar et al. (2021), dividing clients into simple and complex type to train different network architectures collaboratively Acar & Saligrama (2022), and momentum-based updates at the server and client levels to reduce variance Karimireddy et al. (2021); Das et al. (2022). Despite these advancements, FL methods often train a global model that performs well on average across all clients, neglecting individual group performance. This limitation calls for new methods that accommodate groups' demands.

**Clustering and Fairness in Federated Learning.** Preserving privacy by not sharing data or sensitive client information with the server or other clients is a crucial aspect of FL, posing challenges for developing

learning algorithms. Some work Cho et al. (2020); Goetz et al. (2019); Li et al. (2019a) prioritize client selection to boost performance using local validation losses. We demonstrate client discrepancies using the global model's performance at the end of each epoch, though this information is not essential for our algorithm's efficiency. Other approaches Fraboni et al. (2021); Balakrishnan et al. (2022); Jiménez et al. (2024) group the clients based on their representations using similarity matrices or submodular sets. These methods assume a fixed number of clients per round, deviating from FL standards. Approaches like Lyu et al. (2020); Wang et al. (2021) prioritize highly contributing clients, undermining the consensus-based nature of FL. The method in Mohri et al. (2019) minimizes the maximum loss across all data samples to avoid bias towards any data distribution, differing from our server-side debiasing approach using activation vector information. Additionally, Cheng et al. (2024) proposes domain adaptation for client groups with similar characteristics, whereas we focus on disjoint client groups participating in the collaboration. Other works such as Ezzeldin et al. (2023) aim to mitigate the bias between the data samples with a client's data to promote group fairness, whereas we focus on fairness among groups of clients where the goal is to create equitable learning scenarios among different groups. A recent study Yue et al. (2023) also focuses on ensuring fair performance for both groups of clients and individual clients. However, prior knowledge of the client groups is required, while our approach automatically clusters the clients based on the activation vectors. Another concurrent work Chen & Vikalo (2024) orthogonal to ours uses the bias's gradients in the neural network's last layer to construct clusters and then perform heterogeneity-aware client sampling. This work poses an immense computational overhead, requiring computing the similarity matrix amongst all clients to sample the participating clients. Other clustering- based works Vahidian et al. (2023); Sattler et al. (2020) perform clustered federated learning to promote personalization, which is not the objective of our work.

## 3 Background and Preliminaries

In this section, we start by defining the standard terminologies introduced in McMahan et al. (2017); Li et al. (2020) and proceed towards extending the idea to the setting used in our paper.

### 3.1 Federated Learning Setup

FL involves a central server collaborating with $n$ clients, each maintaining its unique data distribution $\mathcal{D}_i$ and sample size $N_i$. The central server aims to train a machine learning model using data from its clients without accessing their local data directly, thus preserving data locality. The expected loss function for client $i$, $\boldsymbol{f}_i(\boldsymbol{w})$, depends on samples $\boldsymbol{x}$ drawn from $\mathcal{D}_i$ and the client's loss function $\ell$, with model parameters $\boldsymbol{w} \in \mathbb{R}^d$. The server minimizes the weighted loss $\boldsymbol{f}(\boldsymbol{w})$ across all clients:

$$\boldsymbol{f}(\boldsymbol{w}) := \sum_{i=1}^{n} p_i \boldsymbol{f}_i(\boldsymbol{w}), \quad \text{where} \quad \boldsymbol{f}_i(\boldsymbol{w}) = \mathbb{E}_{\boldsymbol{x} \sim \mathcal{D}_i}[\ell(\boldsymbol{x}, \boldsymbol{w})], \tag{1}$$

where $p_i = \frac{N_i}{\sum_{i=1}^{n} N_i}$ as suggested by McMahan et al. (2017). Each client performs $\boldsymbol{E}$ local SGD steps per communication round $k \leq \boldsymbol{K}$. In partial client participation, $r \leq n$ clients are randomly selected without replacement in each round. Each client calculates the unbiased stochastic gradient $\widetilde{\nabla} f_i(\boldsymbol{w}; \mathcal{B})$ over a batch $\mathcal{B}$. The FL process iterates through three steps: downlink communication, local computation, and uplink communication, formally described below.

**Downlink Communication**: At the start of each round, the server sends global model parameters to the selected clients:

$$\boldsymbol{w}_{k,0}^i = \boldsymbol{w}_k, \quad \forall i = 1, \dots, n. \tag{2}$$

**Local Computation**: Clients train locally using their datasets:

$$\boldsymbol{w}_{k,\tau+1}^i = \boldsymbol{w}_{k,\tau}^i - \eta_k \widetilde{\nabla} f_i(\boldsymbol{w}_{k,\tau}^i; \mathcal{B}_{k,\tau}^i), \quad \forall \tau = 0, 1, \dots, E-1. \tag{3}$$

Increasing local epochs, especially with non-IID data, causes client drift Li et al. (2020); Karimireddy et al. (2020); Das et al. (2022). To address this, Li et al. (2020) introduced FedProx, adding a proximal term to the update:

$$\boldsymbol{w}_{k,\tau+1}^i = \boldsymbol{w}_{k,\tau}^i - \eta_k \big(\widetilde{\nabla} f_i(\boldsymbol{w}_{k,\tau}^i; \mathcal{B}_{k,\tau}^i) + \mu(\boldsymbol{w}_{k,\tau}^i - \boldsymbol{w}_k)\big), \quad \forall \tau = 0, 1, \dots, E-1. \tag{4}$$

---

**Algorithm 1** `Equitable-FL`

---

1: **Input:** Initial model weights $\boldsymbol{w}_0$, # of communication rounds $K$, period $E$, learning rates $\{\eta_k\}_{k=0}^{K-1}$, and global batch size $r$, # of clusters $C$.
2: **Output:** $\boldsymbol{w}_K$
3: **for** $k = 0, \ldots, K-1$ **do**
4:     Server send $\boldsymbol{w}_k$ to a set of $r$ clients chosen uniformly at random without replacement denoted by $\mathcal{S}_k$,
5:     **for** client $i \in \mathcal{S}_k$ **do**
6:         **Downlink communication:** Set $\boldsymbol{w}_{k,0}^i = \boldsymbol{w}_k$.
7:         **for** $\tau = 0, \ldots, E-1$ **do**
8:             Pick a random batch of samples in client $i$, $\mathcal{B}_{k,\tau}^i$. Compute the stochastic gradient of $f_i$ at $\boldsymbol{w}_{k,\tau}^i$ over $\mathcal{B}_{k,\tau}^i$, viz. $\widetilde{\nabla} f_i(\boldsymbol{w}_{t,\tau}^i; \mathcal{B}_{k,\tau}^i)$.
9:             Update $\boldsymbol{w}_{k,\tau+1}^i = \boldsymbol{w}_{k,\tau}^i - \eta_k\big(\widetilde{\nabla} f_i(\boldsymbol{w}_{k,\tau}^i; \mathcal{B}_{k,\tau}^i) + \mu(\boldsymbol{w}_{k,\tau}^i - \boldsymbol{w}_k)\big)$.
10:         **Uplink communication:** Send $\boldsymbol{w}_{k,E}^i$ and $a_{k,E}^i$.
11:     $S = Similarity\Big(A = \{a_{k,E}^1, a_{k,E}^2, \ldots, a_{k,E}^r\}^\top\Big)$.
12:     $p_k = getprobs(S, C)$.
13:     Update $\boldsymbol{w}_{k+1} = \sum_{i \in \mathcal{S}_k} p_k^i \boldsymbol{w}_{k,E}^i$.
14: **Function:** $Similarity(A)$
15: **Return:** $A \times A^\top$.
16: **Function:** $getprobs(S, C)$
17: Pick the eigen vectors of $S$ corresponding to the $C$ largest eigen values.
18: Use K-Means to cluster the clients.
19: **for** client $i \in \mathcal{S}_k$ **do**
20:     $p_k^i = \frac{1}{C \times \text{\# of clients in each cluster}}$.
21: **Return:** $p_k$.

---

**Uplink Communication**: Clients send their updated parameters to the server:

$$\boldsymbol{w}_{k+1} = \sum_{i \in \mathcal{S}_k} p_i \boldsymbol{w}_{k,E}^i. \tag{5}$$

This iterative process continues for $\boldsymbol{K}$ communication rounds.

## 4 Equitable Clustering

In this section, we present our novel approach named `Equitable-FL`, which aims to tackle the issue of algorithmic bias in FL. Our solution involves implementing a server-side debiasing mechanism that leverages the activation vectors (see Definition 1) to identify and cluster clients based on their similarities. By doing so, we are able to update how the central server aggregates the local model updates received from clients. This approach ensures that the participation across each group of clients is more equitable, minimizing the potential for bias to occur during the learning process.

### 4.1 Problem statement and Motivation

In Federated Learning (FL), the $k$-th communication round's model aggregation involves computing the weighted average of the model updates from each client, given by $\boldsymbol{w}_{k+1} = \sum_{i \in \mathcal{S}_k} p_i \boldsymbol{w}_{k,E}^i$, where $\mathcal{S}_k$ is the set of clients in round $k$. Algorithms such as those proposed by McMahan et al. (2017); Li et al. (2020) weigh clients differently based on factors like the number of data samples, $N_i$, a client possesses. In this scenario, a client's weight is proportional to their data sample size, $p_i = \frac{N_i}{\sum_{i \in \mathcal{S}_k} N_i}$. Alternatively, clients can be weighed uniformly, irrespective of their data sample size, with each client's weight being $\frac{1}{|\mathcal{S}_k|}$. Accurately weighing each client's contribution is crucial for fair and unbiased distributed learning. Simply weighing clients based

on data sample size can lead to biases, favoring clients with more data and giving them disproportionate influence on the global model. This undermines the collaborative nature of FL, which aims to incentivize equal participation. On the other hand, uniformly weighing clients disregards individual contributions, allowing clients with stronger local models to dominate while those with weaker models are suppressed. Both approaches can lead to unfair outcomes, compromising the effectiveness and equity of the learning algorithm.

### 4.2 Framework: Equitable-FL

Our proposed framework, Equitable-FL, addresses these limitations. First, we define what an activation vector is in our context.

**Definition 1** (**Activation vector**). *An activation vector within a neural network refers to the output generated by any given layer after it undergoes transformations like linear combinations (involving inputs, weights, and biases) and activation through a non-linear function. This output captures critical features of the input data, which are vital for the following layers in tasks like classification or prediction. For our purpose, we use the activation vectors, specifically $a_{k,E}^i$, that are generated as outputs of the pre-final layer of the model architecture for $E-th$ local epoch and $k-th$ communication round for client $i$ (See Figure 1).*

At the beginning of the training process, we send the global model $\boldsymbol{w}_k$ to all the clients. Using this global model, each client runs local iterations and transmits the updated model $\boldsymbol{w}_{k+1}^i$ and the activation vector $a_{k,E}^i$ to the central server. The activation vectors are essentially dimensionality-reduced representations of the client's data distribution. Figure 1 describes how we retrieve these $a_{k,E}^i$. The process starts with the **Input**, which moves through several **Deep layers**, including various neural network layers like convolutional, pooling, and fully connected layers. The data then reaches the **Activation layer**, where an activation function, $\phi(\cdot)$ (such as ReLU), applies non-linearity. The activated data continues to the **Classification layer**, which generates the network's final **Output**. Simultaneously, the output from the activation layer is processed through a log softmax function, $\Psi(\cdot)$, to calculate the **Activation vector**. We use these ac-

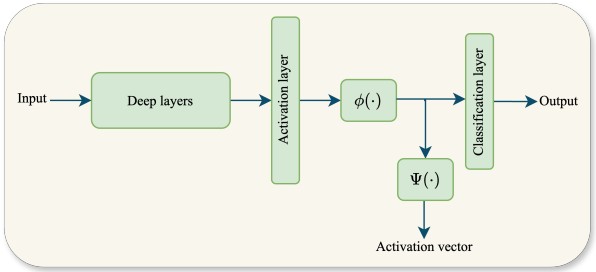

Figure 1: Activation vectors

tivation vectors to construct a similarity matrix $A$ as presented in line 13 of Algorithm 1. Using the K-Means algorithm, this similarity matrix $S$ is then used to perform spectral clustering Von Luxburg (2007). Spectral clustering helps us determine the number of clients belonging to each cluster, which we use to create an equitable client weighing scheme, namely

$$p_k^i = \frac{1}{C \times \# \text{ of clients in each cluster}}. \tag{6}$$

In other words, we use clustering to obtain the weighing probabilities to aggregate the local model for the $k + 1$-th communication round. By doing so, we ensure that all clients are given equal importance and contribute equally to the overall model. So, we formally re-define eq. (1) for our setting as,

$$\boldsymbol{f}(\boldsymbol{w}) := \sum_{q=1}^{C} \frac{1}{|\gamma_q| \times C} \sum_{i \in \gamma_q} \boldsymbol{f}_i(\boldsymbol{w}). \tag{7}$$

**Remark 1.** *Sending the additional $a_{k,E}^i$ to the server does not pose a severe communication cost in comparison to the model parameters because they are mostly of size $\mathcal{O}(10^2)$ in practice, which is considerably smaller than typical model sizes.*

**Remark 2.** *The client weighting scheme $p_{k,E}^i$ described in eq. (6) is straightforward yet effective, as demonstrated in the experiments section, and further is amenable to theoretical analysis. However, one could develop*

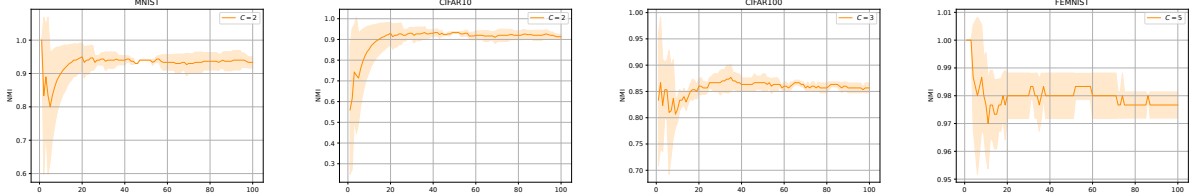

Figure 2: **NMI** comparison across various vision datasets using ResNet-18 model architectures. We partitioned the data among clients to form clusters. The $C$ values in each figure indicate the actual number of clusters we divided the clients into for each dataset. Our observations reveal that the NMIs are close to 1, suggesting that the algorithm's performance aligns closely with the true cluster labels of the clients.

*other weighting strategies based on specific objectives, such as personalization. This paper demonstrates that clients can be clustered effectively by utilizing activation vectors, leading to a fair consensus-based approach. In Figure 2, we present the normalized mutual information (NMI) for various datasets with different cluster sizes and show the effectiveness of our proposed algorithm.*

**Remark 3.** *We emphasize that while K-Means relies on specifying the number of clusters as a hyperparameter, it is not unique in this requirement. Many other clustering algorithms, e.g. hierarchical clustering, also depend on a thresholding mechanism to determine cluster formation, which similarly requires fine-tuning for optimal results. Thus, the need for parameter adjustment is a common aspect across clustering techniques.*

### 4.3 Main Assumptions

In this section, we discuss the standard assumptions that we make in order to provide theoretical guarantees on the convergence of the proposed method.

**Assumption 1** (**Smoothness**)**.** *$\ell(\boldsymbol{x}, \boldsymbol{w})$ is L-smooth with respect to $\boldsymbol{w}$, for all $\boldsymbol{x}$. Thus, each $f_i(\boldsymbol{w})$ $(i \in [n])$ is L-smooth, and so is $f(\boldsymbol{w})$.*

$$\|\nabla f_i(\boldsymbol{w_1}) - \nabla f_i(\boldsymbol{w_2})\| \le L\|\boldsymbol{w_1} - \boldsymbol{w_2}\|; \quad \text{for any } i, \boldsymbol{w_1}, \boldsymbol{w_2}.$$

The assumption stated in Assumption 1 is frequently used while analyzing the convergence of algorithms that employ gradient-based optimization. This assumption has been referenced in several publications such as Chellapandi et al. (2023); Shi et al. (2023); Das et al. (2022). It aims to limit abrupt changes in the gradients.

**Assumption 2** (**Non-negativity**)**.** *Each $f_i(\boldsymbol{w})$ is non-negative and therefore, $f_i^* \triangleq \min f_i(\boldsymbol{w}) \ge 0$.*

The assumption stated in Assumption 2 is usually fulfilled by the loss functions that are employed in practical applications. Nevertheless, if a loss function happens to have a negative value, this assumption can still be met by introducing a constant offset.

**Assumption 3** (**Bounded Variance**)**.** *The variance of the stochastic gradient for all clients $i = 1, \ldots, n$ is bounded, where $\mathcal{B}_{k,\tau}^{(i)}$ represents the random batch of samples in client $i$ for $\tau^{th}$ local iteration.*

$$\mathbb{E}[\|\widetilde{\nabla} f_i(\boldsymbol{w}_{k,\tau}^{(i)}; \mathcal{B}_{k,\tau}^{(i)}) - \nabla f_i(\boldsymbol{w}_{k,\tau}^{(i)})\|^2] \le \sigma^2.$$

Assumption 3 is often utilized to assess the convergence of gradient descent-based algorithms, as demonstrated in various works, including Shi et al. (2023); Li et al. (2019b); Nguyen et al. (2018). However, some other studies assume uniformly bounded stochastic gradients, where $\mathbb{E}[\|\widetilde{\nabla} f_i(\boldsymbol{w}_{k,\tau}^{(i)}; \mathcal{B}_{k,\tau}^{(i)})\|^2] \le \sigma^2$. This assumption is stronger than Assumption 3 and is also shown to be untrue for convex loss functions in Nguyen et al. (2018).

**Assumption 4** (**Existence of Clusters**)**.** *Assuming a system comprising $C$ clusters to which all $n$ clients are allocated, this assumption aligns with the inherent system partitions, for instance, clients segmented*

*by diverse demographic regions. Each cluster, denoted by $\gamma_q$, encapsulates a subset of participants, with $q$ signifying the specific cluster. In the course of the $k^{th}$ communication round, a selection of $r$ clients is made to partake, and they are subsequently distributed into $C$ clusters by our algorithm, ensuring a minimum representation of one client per cluster $q$.*

**Remark 4.** *We ensure a minimum of one participant per cluster for theoretical analysis purposes, though this constraint is relaxed during experimental execution. Notably, Assumption 4 is not a stringent requirement, as the scenario where a cluster lacks participant representation is deemed excessively pessimistic.*

### 4.4 Convergence Analysis

We present the convergence analysis of the proposed Algorithm 1. The detailed proof of Theorem 1 is present in Appendix A.

**Theorem 1** (**Smooth non-convex case of Equitable-FL**). *Suppose Assumptions 1, 2, 3, and 4 hold true for Equitable-FL (refer Algorithm 1). In Equitable-FL set $\eta = \frac{1}{4E\sqrt{3LK}}$. Define a distribution $\mathbb{P}$ for $k \in \{0, \ldots, K-1\}$ such that $\mathbb{P}(k) = \frac{(1+\zeta-1)^{(K-1-k)}}{\sum_{k=0}^{K-1}(1+\zeta)^k}$ where $\zeta := \eta^2 E^2 \left(9\eta L^2 E + 4\eta\mu E + 6L\left(1 + \frac{4\eta^2\mu^2 E^2}{18}\right)\right)$. Sample $k^*$ from $\mathbb{P}$. Then for $\eta LE \leq \frac{1}{2}$, $\eta\mu E \leq \frac{1}{2}$, $\mu < 1$, and $K \geq max\left(\frac{3L}{32}, \frac{\mu^2}{12L}, \frac{\mu^2}{108L^3}\right)$; we have:*

$$\mathbb{E}[\|\nabla f(\boldsymbol{w}_{k^*})\|^2] \leq \frac{16\sqrt{3L}f(\boldsymbol{w}_0)}{(1-\mu)\sqrt{K}} + \frac{\sqrt{L}\sigma^2}{2\sqrt{3K}E(1-\mu)} + \frac{\sigma^2}{36E^2K(1-\mu)} + \frac{(2+E)L\sigma^2}{36K(1-\mu)} + \frac{\mu\sigma^2}{6ELK(1-\mu)}$$
$$+ \frac{\sigma^2}{18CELK(1-\mu)}\left(L^2 + \frac{\mu}{2}\right)\sum_{q=1}^{C}\frac{1}{|\gamma_q|}. \quad (8)$$

*and the expectation is with respect to the randomness in all stochastic gradients and the random selection of $k$ according to the distribution $\mathbb{P}$.*

As stated above, the analysis is conducted without restrictive assumptions such as convexity and bounded client dissimilarity Karimireddy et al. (2020); Upadhyay & Hashemi (2023). Broadly speaking, we can observe that it consists of two terms in eq. (8). In particular, the first term captures the impact of initialization, and the second set of terms results from the noisy stochastic updates of the clients. So, by setting the $\eta$ as described in the theorem, we see that Equitable-FL enjoys a convergence of $\mathcal{O}\left(\frac{1}{\sqrt{K}}\right)$ to reach an $\epsilon$-stationary solution for our setting.

## 5 Experiments

In this section, we present the findings of our framework and compare it with several baselines. We evaluate our algorithm and baselines on an extensive suite of datasets in FL with varying client partition and cluster sizes to show its efficacy.

### 5.1 Datasets and Model Architecture

We conduct deep learning experiments on datasets such as MNIST (Deng, 2012), CIFAR-10, CIFAR-100 (Hinton, 2007), and FEMNIST (Cohen et al., 2017; Caldas et al., 2018). These datasets are standard datasets used in FL experimentation. To showcase the effectiveness of our algorithm, we partition the data in a non-IID fashion. In this partition, we try to emulate the clustering scenario by creating groups among clients and giving them only specific labels. We train a simple CNN model architecture and ResNet-18 (He et al., 2016) for all these datasets. In Table 1, we show how we have planted the clusters. For example, in the case of the MNIST dataset, there are $n = 10$ clients divided into $C = 2$ clusters where $r = 4$ clients are participating in each round. We now describe the data partition and the model architecture for the datasets mentioned above.

**MNIST.** In this case, we train a simple neural network on the MNIST dataset. The total number of clients is $n = 10$, and the data is distributed among these 10 clients. As we described earlier, the division of data

is in a non-IID fashion. Since we do not know the true distribution of data, we created the non-IID and an inherent clustering scenario by distributing the data based on classes. In particular, we ensure that there are two sets of clients where the data possession is entirely orthogonal. So, the first 4 out of the 10 clients get the images from the first 4 out of 10 classes, and the rest classes go to the remaining 6 clients. A client has 800 images of each class, which leads to a client from the first category having 3200 images and a client from the second category having 4800 images, respectively. This approach leads to the formation of two clusters with heterogeneity in terms of the number of samples in each cluster and the nature of samples present in each cluster. From a practical perspective, we tried to emulate the partial client participation scenario. We conducted experiments with 40% of the total participants. The model architecture consists of three fully connected layers, with the first layer accepting flattened input images of size $28 \times 28$ (784 features). The subsequent hidden layers have 200 units each, and ReLU activation functions are applied after the first two layers. The final layer outputs predictions for the classes in the MNIST dataset.

Table 1: Data partition

| Dataset | $C$ | $n$ | $r$ |
|---------|-----|-----|-----|
| MNIST | 2 | 10 | 4 |
| CIFAR-10 | 2 | 10 | 4 |
| CIFAR-100 | 3 | 10 | 4 |
| FEMNIST | 5 | 90 | 18 |

**CIFAR-10.** For this dataset, we train a CNN model. The total number of participants is $n = 10$. The data partition strategy to introduce non-IIDness follows the same strategy as in the case of the MNIST dataset. Similar to the MNIST dataset, we create 2 clusters where the first 4 clients have the data from the first four classes, and the following six clients have the data from the following six classes. So, each client in the first cluster has 3200 images, and the clients from the following cluster have 4800 images each. We experiment with partial client participation where only 40% of the total clients participate. The model architecture is a CNN network with two convolutional layers and three fully connected layers. The first layer is a $5 \times 5$ convolutional layer with 3 and 6 input and output channels, respectively. This is followed by a 5x5 kernel convolutional layer with 16 output channels. A ReLU activation and a max pooling layer succeed each convolutional layer. The resulting output is flattened, traversing two fully connected layers featuring ReLU activations. Finally, the output is directed to the last fully connected layer.

**CIFAR-100.** In CIFAR-100, we follow a similar partition strategy to MNIST and CIFAR-10. The total number of participants is $n = 10$, forming 3 clusters. The first 2 clients have data from the first 20 classes, each having 4400 samples. The second set of 3 clients has the data from the next 30 classes, where each client has 4800 samples, and the third set of 5 clients has the data from the following 50 classes of CIFAR-100, where each client has 5000 samples. In each round, only 40% of the clients participate, thus presenting the partial client participation scenario. The model architecture begins with five convolutional layers, each followed by ReLU activation to introduce non-linearity and three max-pooling layers for downscaling the feature maps. The network concludes with two fully connected layers, with the final layer reducing feature dimensions to 512 and the last layer mapping these to 100 classes, aligning with the CIFAR-100 dataset specifications. The network's forward pass involves processing through these layers, outputting the raw features from the last fully connected layer and the log softmax of these features, catering to feature extraction and classification tasks.

**FEMNIST.** In FEMNIST, we have 5 clusters, and the total number of participants is $n = 90$. The number of clients in each of these clusters is 8, 36, 16, 18, and 12, respectively, and each of these clusters has data from the 2, 8, 15, 20, and 17 classes, respectively. This means the first 8 clients have data from the first 2 classes. The following 36 clients get the data from the following 8 classes, and so on. These numbers are chosen randomly, and there is no correlation between them. In each round, only 20% of the clients participate, thus presenting the partial client participation scenario. This model comprises two convolutional layers, each followed by a max-pooling layer. The convolutional layers use 32 and 64 filters, respectively, with a kernel size of 5. The network also includes a dropout layer with a dropout probability of 0.2 for regularization. The fully connected part of the network consists of two linear layers, with the first linear layer transforming the input from 1024 features to 512 features, followed by ReLu and the final output layer producing 62 classes, corresponding to the EMNIST dataset.

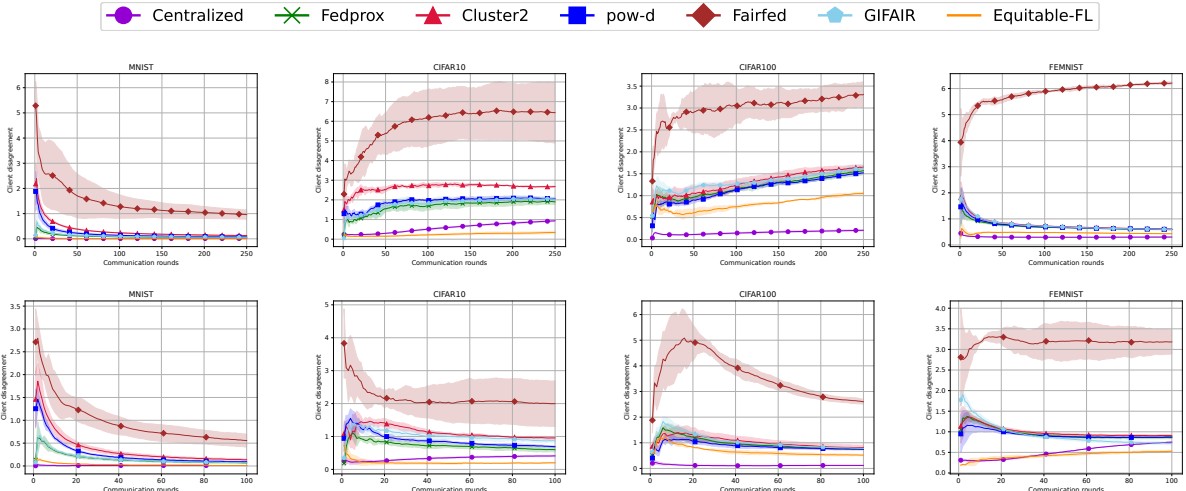

Figure 3: **Client disagreement comparison** on different vision datasets using two different model architectures. The plots in the first row are generated using a simple CNN model architecture, and the plots in the second row are generated using the ResNet-18 model architecture. Equitable-FL consistently outperforms other baselines, invariant to the model architecture, datasets, and number of clusters.

## 5.2 Experiment Setup

**Baselines.** We evaluate our proposed approach against several seminal works in the FL area for tackling client heterogeneity. The baselines include algorithms that tackle client heterogeneity, such as Fedprox Li et al. (2020), and client selection to improve the representation of clients, such as Cluster2 Fraboni et al. (2021). Other algorithms that we compare against propose different model aggregation strategies for groups of clients to promote group fairness, i.e., FairFed Ezzeldin et al. (2023) and GIFAIR-FL-Global Yue et al. (2023). We also compare our method against a biased client selection, pow-d Cho et al. (2020) algorithm that is solely driven by improving accuracy on average. The details are as follows,

- **Centralized**: The training happens in a centralized fashion, where all the clients share their data with the server.

- **Fedprox** (Li et al., 2020): It adds a proximal term to the local client update to mitigate the effect of client drift.

- **Fedprox + Cluster2** (Fraboni et al., 2021): The Cluster2 algorithm uses representative gradients, i.e., the difference between the client's updated model and the global model, to construct the similarity matrix and then sample the clients utilizing it. We use the sampling strategy along with the Fedprox algorithm.

- **Fedprox + pow-d** (Cho et al., 2020): In pow-d the server starts by selecting $d$ clients and then selects a subset of $r$ clients with highest local losses. We use FedProx along with the pow-d algorithm.

- **GIFAIR-FL-Global** (Yue et al., 2023): GIFAIR-FL-Global aims to achieve group fairness by adding the differences in the loss between clusters of clients as a regularization term to the objective function.

- **Fedprox + FairFed (model reweighing)** (Ezzeldin et al., 2023): We utilize the model reweighing strategy proposed in FairFed.

In GIFAIR Yue et al. (2023), it requires an actual number of clients participating from each cluster; thus, it violates the principle of not knowing the client's identity. To conduct the experiments, we provided those details to the algorithm, and in that scenario as well, the algorithm's performance in reducing the

Table 2: **Performance of Algorithms.** We tabulate the accuracy of different algorithms against **Equitable-FL**. The report shows the average test accuracy and $\sigma_{Acc}$ of 3 independent runs over 250 global communication rounds. The results are produced using a simple CNN model architecture. Bold numbers indicate the best results. Additionally, results in the centralized setting row are just for reference; hence, we have not indicated them as best results.

| Method | MNIST | | CIFAR-10 | | CIFAR-100 | | FEMNIST | |
|---|---|---|---|---|---|---|---|---|
| | Acc | $\sigma_{Acc}(\downarrow)$ | Acc | $\sigma_{Acc}(\downarrow)$ | Acc | $\sigma_{Acc}(\downarrow)$ | Acc | $\sigma_{Acc}(\downarrow)$ |
| Centralized | $99.22 \pm 0.03$ | $0.17 \pm 0.03$ | $58.74 \pm 0.75$ | $7.18 \pm 0.28$ | $47.01 \pm 0.18$ | $2.27 \pm 0.16$ | $91.18 \pm 0.015$ | $11.70 \pm 0.08$ |
| Fedprox | $96.97 \pm 0.20$ | $2.26 \pm 0.21$ | $47.36 \pm 0.50$ | $18.01 \pm 0.68$ | $33.04 \pm 0.14$ | $13.90 \pm 0.43$ | $74.18 \pm 0.12$ | $22.99 \pm 0.08$ |
| Cluster2 | $95.38 \pm 0.18$ | $4.51 \pm 0.23$ | $47.17 \pm 0.80$ | $25.52 \pm 0.38$ | $32.66 \pm 0.13$ | $15.37 \pm 0.21$ | $\mathbf{76.12} \pm 0.01$ | $21.37 \pm 0.13$ |
| Pow-d | $96.03 \pm 0.33$ | $3.61 \pm 0.40$ | $47.08 \pm 0.50$ | $22.04 \pm 0.68$ | $33.07 \pm 0.41$ | $14.50 \pm 0.22$ | $74.34 \pm 0.14$ | $22.80 \pm 0.13$ |
| FairFed | $82.10 \pm 1.75$ | $20.2 \pm 2.12$ | $44.74 \pm 0.84$ | $33.52 \pm 3.01$ | $26.79 \pm 1.63$ | $17.57 \pm 1.80$ | $54.41 \pm 0.61$ | $41.56 \pm 1.05$ |
| GIFAIR | $96.98 \pm 0.12$ | $2.34 \pm 0.13$ | $47.41 \pm 0.83$ | $18.88 \pm 0.98$ | $33.26 \pm 0.43$ | $14.28 \pm 0.48$ | $74.13 \pm 0.12$ | $22.96 \pm 0.25$ |
| Equitable-FL | $\mathbf{97.75} \pm 0.20$ | $\mathbf{1.17} \pm 0.25$ | $\mathbf{49.74} \pm 0.56$ | $\mathbf{6.07} \pm 0.31$ | $\mathbf{33.34} \pm 0.10$ | $\mathbf{10.57} \pm 0.13$ | $75.54 \pm 0.25$ | $\mathbf{16.21} \pm 0.04$ |

Table 3: **Performance of Algorithms.** We tabulate the accuracy of different algorithms against **Equitable-FL**. The report shows the average test accuracy and $\sigma_{Acc}$ of 3 independent runs over 100 global communication rounds. The results are produced using the ResNet-18 model architecture. Bold numbers indicate the best results. Additionally, results in the centralized setting row are just for reference; hence, we have not indicated them as the best results.

| Method | MNIST | | CIFAR-10 | | CIFAR-100 | | FEMNIST | |
|---|---|---|---|---|---|---|---|---|
| | Acc | $\sigma_{Acc}(\downarrow)$ | Acc | $\sigma_{Acc}(\downarrow)$ | Acc | $\sigma_{Acc}(\downarrow)$ | Acc | $\sigma_{Acc}(\downarrow)$ |
| Centralized | $99.43 \pm 0.02$ | $0.10 \pm 0.02$ | $83.60 \pm 0.10$ | $5.41 \pm 0.21$ | $65.90 \pm 0.21$ | $0.10 \pm 0.16$ | $90.51 \pm 0.07$ | $11.67 \pm 0.01$ |
| Fedprox | $96.01 \pm 0.52$ | $2.84 \pm 0.48$ | $66.72 \pm 0.80$ | $15.70 \pm 1.36$ | $55.19 \pm 0.11$ | $16.85 \pm 0.23$ | $67.61 \pm 1.07$ | $30.29 \pm 0.35$ |
| Cluster2 | $94.53 \pm 0.10$ | $5.20 \pm 0.03$ | $63.46 \pm 0.59$ | $26.61 \pm 0.70$ | $53.99 \pm 0.83$ | $18.71 \pm 1.25$ | $68.12 \pm 0.76$ | $30.27 \pm 0.54$ |
| Pow-d | $95.05 \pm 0.46$ | $4.55 \pm 0.73$ | $65.06 \pm 1.25$ | $21.48 \pm 1.85$ | $54.91 \pm 0.39$ | $17.53 \pm 0.65$ | $67.67 \pm 0.20$ | $30.02 \pm 0.13$ |
| FairFed | $85.85 \pm 2.92$ | $15.41 \pm 3.81$ | $57.10 \pm 2.11$ | $31.98 \pm 7.50$ | $46.05 \pm 0.48$ | $24.98 \pm 0.18$ | $53.27 \pm 1.344$ | $38.92 \pm 2.22$ |
| GIFAIR | $96.02 \pm 0.33$ | $2.86 \pm 0.31$ | $64.41 \pm 0.56$ | $20.25 \pm 1.21$ | $55.39 \pm 0.14$ | $17.08 \pm 0.13$ | $\mathbf{69.55} \pm 0.62$ | $25.53 \pm 0.61$ |
| Equitable-FL | $\mathbf{96.95} \pm 0.40$ | $\mathbf{1.65} \pm 0.53$ | $\mathbf{69.40} \pm 0.48$ | $\mathbf{7.83} \pm 0.71$ | $\mathbf{55.65} \pm 0.15$ | $\mathbf{13.67} \pm 0.39$ | $67.02 \pm 0.42$ | $\mathbf{24.11} \pm 0.33$ |

disagreement between clients is inferior to ours. Moreover, in Fairfed Ezzeldin et al. (2023), we only utilize their model/client weighing scheme, and it turns out that their strategy can generate negative weights (refer to the section on Computing Aggregation Weights for FairFed in Ezzeldin et al. (2023)). So, to mitigate the issue, we lower-bound these aggregation weights to 0.

**Implementation details.** We implement all algorithms in PyTorch using an Nvidia A100 GPU. We make Assumption 4 solely for convergence analysis, not for experimental evaluations. The figures and tables present results averaged over three independent runs. Following the standard FL learning regime, we start with a global communication round of $K = 250$ for a simple CNN architecture and $K = 100$ for ResNet-18, with local epochs set to $E = 5$ unless stated otherwise. We perform hyperparameter tuning with $\eta \in \{0.001, 0.01, 0.1\}$ and $\mu \in \{0.001, 0.01, 0.1, 1\}$, and present results for the best outcomes. We use the cumulative moving average of the results in our graphical representations to enhance clarity.

**Evaluation metric.** To demonstrate the effectiveness of our algorithm, we measure client disagreement using test loss. Client disagreement is based on the principle that the global model should perform equally well on each client's dataset to mitigate algorithmic bias. Specifically, we define client disagreement as the average absolute difference in loss values between two participating clients during a communication round using the updated global model, expressed as:

$$CD_{k+1} = \frac{\sum_{i \in \mathcal{S}_k} \sum_{j \in \mathcal{S}_k} |\boldsymbol{f}_i(\boldsymbol{w}_{k+1}) - \boldsymbol{f}_j(\boldsymbol{w}_{k+1})|}{\binom{r}{2}}. \tag{9}$$

As discussed, $CD$ assesses algorithm performance and efficiency. It is not essential for the algorithm's function but is a useful evaluation tool. Additionally, we evaluate the standard deviation between the global test accuracy and each client's accuracy, denoted as $\sigma_{Acc}$, as an additional performance metric to measure client performance discrepancy (refer to Yue et al. (2023)).

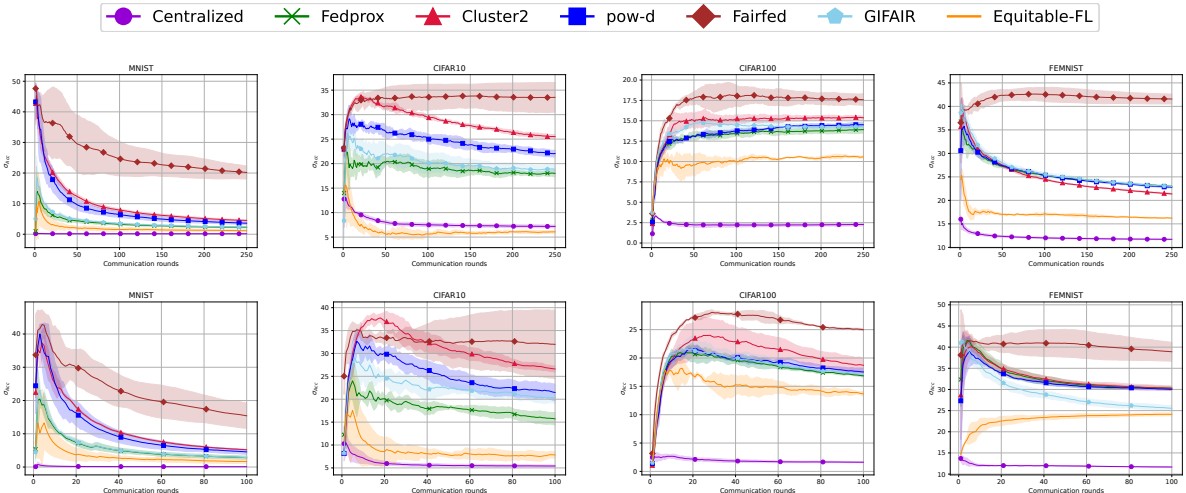

Figure 4: $\sigma_{\mathbf{Acc}}$ **comparison** on different vision datasets using two different model architectures. The plots in the first row are generated using simple CNN model architecture, and the plots in the second row are generated using the ResNet-18 model architecture. Equitable-FL consistently outperforms other baselines, invariant to the model architecture, datasets, and number of clusters.

## 5.3 Main Results

In this study, we evaluate the performance of our algorithm against established baselines. To ensure comprehensive results, we conducted experiments on various datasets, including the widely used MNIST, CIFAR-10, CIFAR-100, and FEMNIST datasets. The datasets are distributed among clients such that the clients form clusters. Our findings demonstrate that Equitable-FL consistently outperforms other baselines in reducing client disagreement in highly heterogeneous settings, as illustrated in Figure 3. Additionally, we show that the effectiveness of our framework in mitigating bias is independent of the model architecture used. The average accuracy and $\sigma_{Acc}$ are presented in Tables 2 and 3 as well as in Figures 4 and 5. Our results indicate that the Equitable-FL significantly outperforms other baselines, except the FEMNIST dataset (regarding average accuracy). Overall, our results suggest that Equitable-FL is a promising solution for addressing challenges associated with distributed machine learning.

Table 4: We tabulate the accuracy and $\sigma_{Acc}$ of our algorithm using the CIFAR-100 dataset for ResNet-18 with varying $C$ and $E$. The table shows the average test accuracy of 3 independent runs over 50 global communication rounds. Bold numbers indicate the best results.

| Method | $C = 2$ | | $C = 3$ | | $C = 4$ | |
|---|---|---|---|---|---|---|
| | Acc | $\sigma_{Acc}(\downarrow)$ | Acc | $\sigma_{Acc}(\downarrow)$ | Acc | $\sigma_{Acc}(\downarrow)$ |
| Equitable-FL | $49.71 \pm 1.16$ | $18.02 \pm 1.16$ | $\mathbf{49.89 \pm 0.66}$ | $\mathbf{14.92 \pm 1.72}$ | $49.13 \pm 0.63$ | $19.04 \pm 0.80$ |
| Method | $E = 1$ | | $E = 5$ | | $E = 10$ | |
| | Acc | $\sigma_{Acc}(\downarrow)$ | Acc | $\sigma_{Acc}(\downarrow)$ | Acc | $\sigma_{Acc}(\downarrow)$ |
| Equitable-FL | $31.93 \pm 0.78$ | $13.47 \pm 1.50$ | $49.90 \pm 0.65$ | $14.92 \pm 1.72$ | $\mathbf{52.90 \pm 0.67}$ | $\mathbf{14.45 \pm 1.60}$ |

## 5.4 Abalation study

We evaluate our proposed framework on the CIFAR-100 dataset using the ResNet-18 architecture. During the experiment, we vary the number of clusters, $C$, the number of local epochs, $E$, and the proximal term, $\mu$. In Table 4, we show that with correct cluster assignment, i.e., $C = 3$, our framework significantly reduces the disagreement among clients as well as improves the test accuracy. Additionally, we show in Table 4 that with increasing local iterations, our algorithm consistently manages to reduce the disagreement among

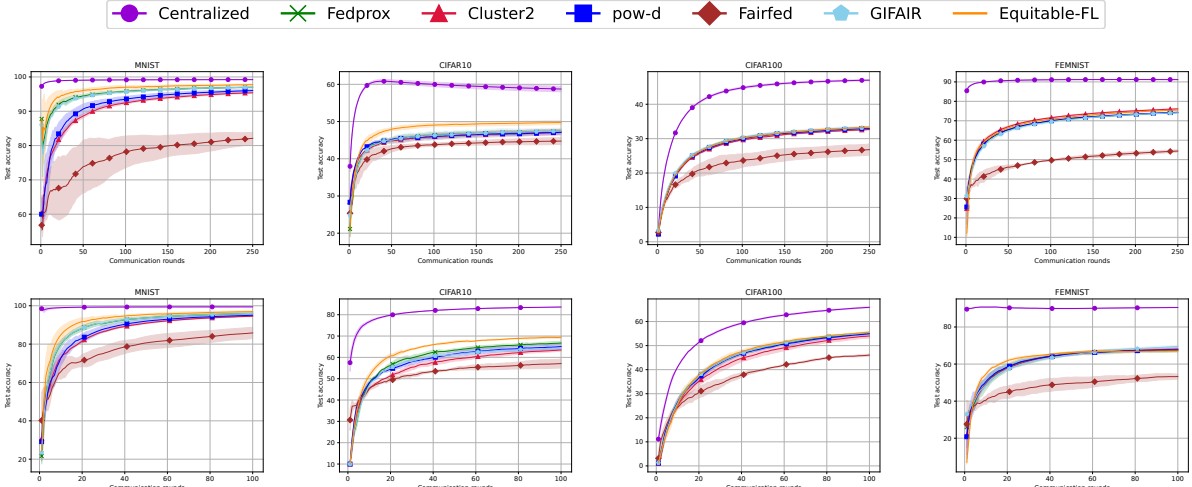

Figure 5: **Test accuracy comparison** on different vision datasets using two different model architectures. The plots in the first row are generated using the model architecture described in section 5.1, and the plots in the second row are generated using the ResNet-18 model architecture. Equitable-FL consistently outperforms other baselines, invariant to the model architecture, datasets, and number of clusters.

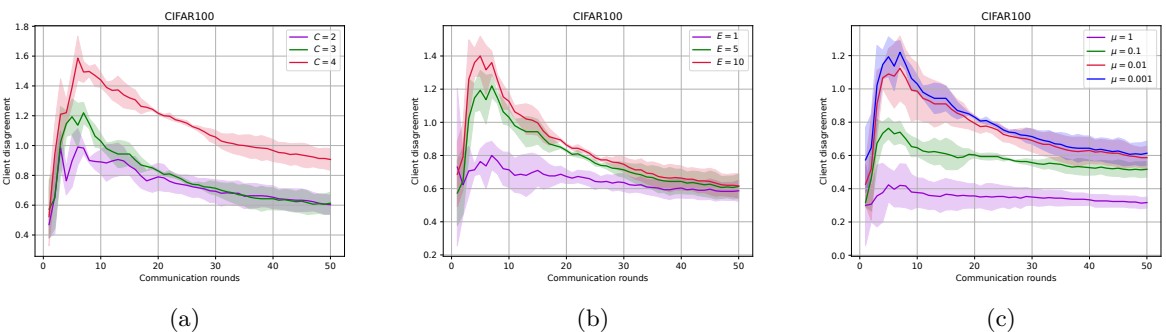

Figure 6: **Impact of hyperparameters on client disagreement:** perturbations in $C$, $E$ and $\mu$.

clients, mitigating the effect of client drift. In Table 5, we show that if we increase the $\mu$, the disagreement or the effect of client drift will reduce but at the price of accuracy. In Figure 6, we show the $CD$ comparison for these 3 cases.

Table 5: We tabulate the accuracy and $\sigma_{Acc}$ of our algorithm using the CIFAR-100 dataset for ResNet-18 with varying $\mu$. The table shows the average test accuracy of 3 independent runs over 50 global communication rounds. Bold numbers indicate the best results.

| Method | $\mu = 0.001$ | | $\mu = 0.01$ | | $\mu = 0.1$ | | $\mu = 1$ | |
|---|---|---|---|---|---|---|---|---|
| | Acc | $\sigma_{Acc}(\downarrow)$ | Acc | $\sigma_{Acc}(\downarrow)$ | Acc | $\sigma_{Acc}(\downarrow)$ | Acc | $\sigma_{Acc}(\downarrow)$ |
| Equitable-FL | $\mathbf{49.89} \pm 0.65$ | $14.92 \pm 1.72$ | $49.44 \pm 0.23$ | $15.06 \pm 1.83$ | $35.18 \pm 0.61$ | $12.92 \pm 1.27$ | $5.24 \pm 0.07$ | $\mathbf{4.32} \pm 0.30$ |

# 6 Conclusion and Future Work

In this paper, we presented an equitable learning framework for FL, which reduces the bias against a diverse set of participants. We utilize the side information offered by the activation vectors to cluster the clients into groups based on their similarity and use this to propose a weighing mechanism that promotes fairness.

Additionally, we established a rate of convergence to reach a stationary solution for Equitable-FL. We visualized the efficacy of our proposed framework on various vision datasets and showed that it consistently outperforms the baseline in mitigating bias. As previously mentioned, privacy is a cornerstone of Federated Learning (FL). However, activation vectors can lead to information leakage if the server is compromised. Despite this risk, privacy-preserving mechanisms exist to enable secure learning without compromising privacy Schaffer et al. (2012); Qiao et al. (2024). This paper aims to demonstrate the effectiveness of using activation vectors to cluster client groups and develop a fair weighting mechanism for these groups. We also propose that this clustering approach can be applied to personalized federated learning Long et al. (2023); Ghosh et al. (2020). This method effectively clusters client groups and ensures fair solutions for each group, maintaining fairness for all participants.

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

# Appendix

## A Proof of Theorem 1

We base our convergence analysis on a framework similar to that of Das et al. (2022). However, our approach introduces distinct variations due to the addition of a proximal term in local updates and the equitable reweighting of these updates across clients, leading to differences in the analysis compared to Das et al. (2022).

*Proof.* From Lemma 1, for $\eta_k LE \leq \frac{1}{2}$, $\eta_k \mu E \leq \frac{1}{2}$, we upper bound the per-round progress as:

$$\mathbb{E}\left[f(\boldsymbol{w}_{k+1})\right] \leq \mathbb{E}\left[f(\boldsymbol{w}_k)\right] - \frac{\eta_k E(1-\mu)}{2} \mathbb{E}\left[\|\nabla f(\boldsymbol{w}_k)\|^2\right]$$

$$+ \eta_k^2 E^2 \left(9\eta_k L^2 E + 4\eta_k \mu E + 6L\left(1 + \frac{4\eta_k^2 \mu^2 E^2}{18}\right)\right) \sum_{q=1}^C \frac{1}{|\gamma_q| \times C} \sum_{i \in \gamma_q} \mathbb{E}\left[\|\nabla f_i(\boldsymbol{w}_k)\|^2\right]$$

$$+ \eta_k^2 E \left(L\left(1 + \frac{2\eta_k}{3}\right) + \frac{8\eta_k L^2 E(2+E)}{9} + \frac{4\eta_k \mu E(1 + \eta_k \mu E)}{9}\right.$$

$$\left. + \frac{4\eta_k E}{3C}\left(L^2 + \frac{\mu}{2}\right) \sum_{q=1}^C \frac{1}{|\gamma_q|}\right) \sigma^2. \quad (10)$$

Now using $L$-smoothness and 2 of $f_i$'s, we get:

$$\sum_{q=1}^C \frac{1}{|\gamma_q| \times C} \sum_{i \in [\gamma_q]} \mathbb{E}[\|\nabla f_i(\boldsymbol{w}_k)\|^2] \leq \sum_{q=1}^C \frac{2L}{|\gamma_q| \times C} \sum_{i \in [\gamma_q]} \left(\mathbb{E}[f_i(\boldsymbol{w}_k)] - f_i^*\right)$$

$$\sum_{q=1}^C \frac{1}{|\gamma_q| \times C} \sum_{i \in [\gamma_q]} \mathbb{E}[\|\nabla f_i(\boldsymbol{w}_k)\|^2] \leq 2L\mathbb{E}[f(\boldsymbol{w}_k)] - \sum_{q=1}^C \frac{2L}{|\gamma_q| \times C} \sum_{i \in [\gamma_q]} f_i^*$$

$$\sum_{q=1}^C \frac{1}{|\gamma_q| \times C} \sum_{i \in [\gamma_q]} \mathbb{E}[\|\nabla f_i(\boldsymbol{w}_k)\|^2] \leq 2nL\mathbb{E}[f(\boldsymbol{w}_k)]. \quad (11)$$

Using eq. (11) in eq. (10), we get for a constant learning rate of $\eta_k = \eta$:

$$\mathbb{E}\left[f(\boldsymbol{w}_{k+1})\right] \leq \left(1 + \eta^2 E^2\left(9\eta L^2 E + 4\eta \mu E + 6L\left(1 + \frac{4\eta^2 \mu^2 E^2}{18}\right)\right)\right) \mathbb{E}\left[f(\boldsymbol{w}_k)\right] - \frac{\eta E(1-\mu)}{2} \mathbb{E}\left[\|\nabla f(\boldsymbol{w}_k)\|^2\right]$$

$$+ \eta^2 E\left(L\left(1 + \frac{2\eta}{3}\right) + \frac{8\eta L^2 E(2+E)}{9} + \frac{4\eta \mu E(1 + \eta \mu E)}{9} + \frac{4\eta E}{3C}\left(L^2 + \frac{\mu}{2}\right) \sum_{q=1}^C \frac{1}{|\gamma_q|}\right) \sigma^2. \quad (12)$$

For clarity, define $\zeta := \eta^2 E^2 \left(9\eta L^2 E + 4\eta \mu E + 6L\left(1 + \frac{4\eta^2 \mu^2 E^2}{18}\right)\right)$ and

$$\zeta_2 := \eta^2 E\left(L\left(1 + \frac{2\eta}{3}\right) + \frac{8\eta L^2 E(2+E)}{9} + \frac{4\eta \mu E(1 + \eta \mu E)}{9} + \frac{4\eta E}{3C}\left(L^2 + \frac{\mu}{2}\right) \sum_{q=1}^C \frac{1}{|\gamma_q|}\right).$$

Then unfolding the recursion from $k = 0$ to $k = K - 1$, we get:

$$\mathbb{E}\left[f(\boldsymbol{w}_K)\right] \leq (1 + \zeta_1)^K \mathbb{E}\left[f(\boldsymbol{w}_k)\right] - \frac{\eta E(1-\mu)}{2} \sum_{k=0}^{K-1} (1+\zeta)^{(K-1-k)} \mathbb{E}\left[\|\nabla f(\boldsymbol{w}_k)\|^2\right]$$

$$+ \eta^2 E \zeta_2 \sigma^2 \sum_{k=0}^{K-1} (1+\zeta)^{(K-1-k)}. \quad (13)$$

Let us define $p_k := \frac{(1+\zeta)^{(K-1-k)}}{\sum_{k'=0}^{K-1}(1+\zeta_1)^{(K-1-k')}}$. Then, set $\mu < 1$ and re-arranging eq. (13) using the fact that $\mathbb{E}[f(\boldsymbol{w}_K)] \geq 0$, we get:

$$\sum_{k=0}^{K-1} p_k \mathbb{E}[\|\nabla f(\boldsymbol{w}_k)\|^2] \leq \frac{2(1+\zeta)^K f(\boldsymbol{w}_0)}{\eta E(1-\mu)\sum_{k'=0}^{K-1}(1+\zeta)^{k'}} + \frac{2\eta\zeta_2}{(1-\mu)}\sigma^2 \tag{14}$$

$$= \frac{2\zeta f(\boldsymbol{w}_0)}{\eta E(1-\mu)\left(1-(1+\zeta)^{-K}\right)} + \frac{2\eta\zeta_2}{(1-\mu)}\sigma^2, \tag{15}$$

where the eq. (15) follows by using the fact that $\sum_{k'=0}^{K-1}(1+\zeta)^{k'} = \frac{(1+\zeta)^K - 1}{\zeta}$. Now,

$$(1+\zeta_1)^{-K} < 1 - \zeta K + \zeta^2 \frac{K(K+1)}{2} < 1 - \zeta K + \zeta^2 K^2$$

$$\implies 1 - (1+\zeta)^{-K} > \zeta K(1 - \zeta K).$$

Plugging this in eq. (15), we have for $\zeta K < 1$:

$$\sum_{k=0}^{K-1} p_k \mathbb{E}[\|\nabla f(\boldsymbol{w}_k)\|^2] \leq \frac{2f(\boldsymbol{w}_0)}{\eta E K(1-\mu)(1-\zeta K)} + \frac{2\eta\zeta_2}{(1-\mu)}\sigma^2. \tag{16}$$

Now, note that the optimal step size will be $\eta = \mathcal{O}\left(\frac{1}{E\sqrt{LK}}\right)$. So, then let us pick $\eta = \frac{1}{4E\sqrt{3LK}}$. We need to have $\eta LE \leq \frac{1}{2}$ and $\eta \mu E \leq \frac{1}{2}$; which happens for $K \geq max\left(\frac{L}{12}, \frac{\mu^2}{12L}\right)$. Furthermore, lets ensure that $\zeta K < \frac{1}{2}$; which happens for $K \geq max\left(\frac{3L}{32}, \frac{\mu^2}{108L^3}, \frac{\mu^2}{72L}\right)$. Therefore we should have $K \geq max\left(\frac{3L}{32}, \frac{\mu^2}{12L}, \frac{\mu^2}{108L^3}\right)$. So, plugging in $\eta = \frac{1}{4E\sqrt{3LK}}$ and $1 - \zeta K \geq \frac{1}{2}$ in eq. (16); we get,

$$\sum_{k=0}^{K-1} p_k \mathbb{E}[\|\nabla f(\boldsymbol{w}_k)\|^2] \leq \frac{16\sqrt{3L}f(\boldsymbol{w}_0)}{(1-\mu)\sqrt{K}} + \frac{\sqrt{L}\sigma^2}{2\sqrt{3K}E(1-\mu)}$$

$$+ \frac{\sigma^2}{36E^2K(1-\mu)} + \frac{(2+E)L\sigma^2}{36K(1-\mu)} + \frac{\mu\sigma^2}{6ELK(1-\mu)}$$

$$+ \frac{\sigma^2}{18CELK(1-\mu)}\left(L^2 + \frac{\mu}{2}\right)\sum_{q=1}^{C}\frac{1}{|\gamma_q|}. \tag{17}$$

This finishes the proof. $\qquad\square$

## B  Supporting Lemmas

**Lemma 1.** *For $\eta_k LE \leq \frac{1}{2}$ and For $\eta_k \mu E \leq \frac{1}{2}$,*

$$\mathbb{E}\left[f(\boldsymbol{w}_{k+1})\right] \leq \mathbb{E}\left[f(\boldsymbol{w}_k)\right] - \frac{\eta_k E(1-\mu)}{2}\mathbb{E}\left[\|\nabla f(\boldsymbol{w}_k)\|^2\right]$$

$$+ \eta_k^2 E^2\left(9\eta_k L^2 E + 4\eta_k \mu E + 6L\left(1 + \frac{4\eta_k^2\mu^2 E^2}{18}\right)\right)\sum_{q=1}^{C}\frac{1}{|\gamma_q| \times C}\sum_{i\in\gamma_q}\mathbb{E}\left[\|\nabla f_i(\boldsymbol{w}_k)\|^2\right]$$

$$+ \eta_k^2 E\left(L\left(1 + \frac{2\eta_k}{3}\right) + \frac{8\eta_k L^2 E(2+E)}{9} + \frac{4\eta_k \mu E(1+\eta_k\mu E)}{9}\right.$$

$$\left. + \frac{4\eta_k E}{3C}\left(L^2 + \frac{\mu}{2}\right)\sum_{q=1}^{C}\frac{1}{|\gamma_q|}\right)\sigma^2. \tag{18}$$

*Proof.* Define

$$\boldsymbol{w}_{k,\tau}^i = \boldsymbol{w}_k - \eta_k \sum_{t=0}^{\tau-1} \widetilde{\nabla} f_i \left( \boldsymbol{w}_{k,t}^i; \mathcal{B}_{k,t}^i \right) - \eta_k \mu \sum_{t=0}^{\tau-1} \left( \boldsymbol{w}_{k,t}^i - \boldsymbol{w}_k \right), \tag{19}$$

$$\overline{\boldsymbol{w}}_{k,\tau} = \boldsymbol{w}_k - \eta_k \sum_{q=1}^{C} \frac{1}{|\gamma_q| \times C} \sum_{i \in \gamma_q} \sum_{t=0}^{\tau-1} \widetilde{\nabla} f_i \left( \boldsymbol{w}_{k,t}^i; \mathcal{B}_{k,t}^i \right) - \eta_k \mu \sum_{q=1}^{C} \frac{1}{|\gamma_q| \times C} \sum_{i \in \gamma_q} \sum_{t=0}^{\tau-1} \left( \boldsymbol{w}_{k,t}^i - \boldsymbol{w}_k \right), \tag{20}$$

$$\boldsymbol{w}_{k+1} = \boldsymbol{w}_k - \eta_k \sum_{q=1}^{C} \frac{1}{|\gamma_{q_k}| \times C} \sum_{i \in \gamma_{q_k}} \sum_{\tau=0}^{E-1} \widetilde{\nabla} f_i \left( \boldsymbol{w}_{k,\tau}^i; \mathcal{B}_{k,\tau}^i \right)$$
$$- \eta_k \mu \sum_{q=1}^{C} \frac{1}{|\gamma_{q_k}| \times C} \sum_{i \in \gamma_{q_k}} \sum_{\tau=0}^{E-1} \left( \boldsymbol{w}_{k,\tau}^i - \boldsymbol{w}_k \right). \tag{21}$$

For any two vectors $\boldsymbol{a}$ and $\boldsymbol{b}$, we have

$$\langle \boldsymbol{a}, \boldsymbol{b} \rangle = \frac{1}{2} \left( \|\boldsymbol{a}\|^2 + \|\boldsymbol{b}\|^2 - \|\boldsymbol{a} - \boldsymbol{b}\|^2 \right). \tag{22}$$

Also, note

$$\mathbb{E}_{\{\mathcal{B}_{k,t}^i\}_{i=1}^{|\gamma_q|}} \left[ \frac{1}{|\gamma_q|} \sum_{i \in \gamma_q} \widetilde{\nabla} f_i \left( \boldsymbol{w}_{k,t}^i; \mathcal{B}_{k,t}^i \right) \right] = \frac{1}{|\gamma_q|} \sum_{i \in \gamma_q} \nabla f_i \left( \boldsymbol{w}_{k,t}^i \right), \tag{23}$$

$$\mathbb{E}_{\{\mathcal{B}_{k,t}^i\}_{i=1,t=0}^{|\gamma_q|,\tau-1}} \left[ \left\| \sum_{t=0}^{\tau-1} \frac{1}{|\gamma_q|} \sum_{i \in \gamma_q} \widetilde{\nabla} f_i \left( \boldsymbol{w}_{k,t}^i; \mathcal{B}_{k,t}^i \right) \right\|^2 \right] = \tau \sum_{t=0}^{\tau-1} \mathbb{E} \left[ \left\| \frac{1}{|\gamma_q|} \sum_{i \in \gamma_q} \nabla f_i \left( \boldsymbol{w}_{k,t}^i \right) \right\|^2 \right] + \frac{\tau \sigma^2}{|\gamma_q|}, \tag{24}$$

$$\mathbb{E}_{\{\mathcal{B}_{k,t}^i\}_{t=0}^{\tau-1}} \left[ \left\| \sum_{t=0}^{\tau-1} \widetilde{\nabla} f_i \left( \boldsymbol{w}_{k,t}^i; \mathcal{B}_{k,t}^i \right) \right\|^2 \right] = \tau \sum_{t=0}^{\tau-1} \mathbb{E} \left[ \| \nabla f_i \left( \boldsymbol{w}_{k,t}^i \right) \|^2 \right] + \tau \sigma^2. \tag{25}$$

Since $\sigma^2$ is the maximum variance of the stochastic gradients, eqs. (24) and (25) follow due to the independence of the noise in each local update of each client. Now using the $L$-smoothness of $f$ and eq. (21), we get

$$\mathbb{E}[f(\boldsymbol{w}_{k+1})] \leq \mathbb{E}[f(\boldsymbol{w}_k)] + (A) + (B) + (M), \tag{26}$$

where

$$A = -\eta_k \mathbb{E} \left[ \left\langle \nabla f(\boldsymbol{w}_k), \sum_{q=1}^{C} \frac{1}{|\gamma_{q_k}| \times C} \sum_{i \in \gamma_{q_k}} \sum_{\tau=0}^{E-1} \widetilde{\nabla} f_i \left( \boldsymbol{w}_{k,t}^i; \mathcal{B}_{k,\tau}^i \right) \right\rangle \right],$$

$$B = -\eta_k \mu \mathbb{E} \left[ \left\langle \nabla f(\boldsymbol{w}_k), \sum_{q=1}^{C} \frac{1}{|\gamma_{q_k}| \times C} \sum_{i \in \gamma_{q_k}} \sum_{\tau=0}^{E-1} \left( \boldsymbol{w}_{k,\tau}^i - \boldsymbol{w}_k \right) \right\rangle \right],$$

and

$$M = \frac{\eta_k^2 L}{2} \mathbb{E} \left[ \left\| \sum_{q=1}^{C} \frac{1}{|\gamma_{q_k}| \times C} \sum_{i \in \gamma_{q_k}} \sum_{\tau=0}^{E-1} \widetilde{\nabla} f_i \left( \boldsymbol{w}_{k,\tau}^i; \mathcal{B}_{k,\tau}^i \right) + \mu \sum_{q=1}^{C} \frac{1}{|\gamma_{q_k}| \times C} \sum_{i \in \gamma_{q_k}} \sum_{\tau=0}^{E-1} \left( \boldsymbol{w}_{k,\tau}^i - \boldsymbol{w}_k \right) \right\|^2 \right].$$

Starting with (A) by taking an expectation over the set $\gamma_{q_k}$,

$$
\begin{aligned}
A &= -\eta_k \sum_{\tau=0}^{E-1} \mathbb{E}\left[\left\langle \nabla f(\boldsymbol{w}_k), \sum_{q=1}^{C} \frac{1}{|\gamma_q| \times C} \sum_{i \in \gamma_q} \widetilde{\nabla} f_i\left(\boldsymbol{w}_{k,t}^i; \mathcal{B}_{k,\tau}^i\right) \right\rangle\right] \\
&= -\frac{\eta_k E}{2} \mathbb{E}\left[\|\nabla f(\boldsymbol{w}_k)\|^2\right] - \frac{\eta_k}{2} \sum_{\tau=0}^{E-1} \mathbb{E}\left[\left\|\sum_{q=1}^{C} \frac{1}{|\gamma_q| \times C} \sum_{i \in \gamma_q} \nabla f_i\left(\boldsymbol{w}_{k,\tau}^i\right)\right\|^2\right] \\
&\quad + \frac{\eta_k}{2} \sum_{\tau=0}^{E-1} \mathbb{E}\left[\left\|\nabla f(\boldsymbol{w}_k) - \sum_{q=1}^{C} \frac{1}{|\gamma_q| \times C} \sum_{i \in \gamma_q} \nabla f_i\left(\boldsymbol{w}_{k,\tau}^i\right)\right\|^2\right] \quad (27) \\
&\leq -\frac{\eta_k E}{2} \mathbb{E}\left[\|\nabla f(\boldsymbol{w}_k)\|^2\right] + \underbrace{\frac{\eta_k}{2} \sum_{\tau=0}^{E-1} \mathbb{E}\left[\left\|\nabla f(\boldsymbol{w}_k) - \sum_{q=1}^{C} \frac{1}{|\gamma_q| \times C} \sum_{i \in \gamma_q} \nabla f_i\left(\boldsymbol{w}_{k,\tau}^i\right)\right\|^2\right]}_{A_1}. \quad (28)
\end{aligned}
$$

Note eq. (27) follows due to eq. (23), linearity of inner product and eq. (22) while eq. (28) follows by dropping the second term on the right side of eq. (27). Now using $A_1$,

$$
\begin{aligned}
A_1 &= \frac{\eta_k}{2} \sum_{\tau=0}^{E-1} \mathbb{E}\left[\left\|\nabla f(\boldsymbol{w}_k) - \sum_{q=1}^{C} \frac{1}{|\gamma_q| \times C} \sum_{i \in \gamma_q} \nabla f_i\left(\boldsymbol{w}_{k,\tau}^i\right) - \nabla f(\overline{\boldsymbol{w}}_{k,\tau}) + \nabla f(\overline{\boldsymbol{w}}_{k,\tau})\right\|^2\right] \\
&\leq \eta_k \sum_{\tau=0}^{E-1} \mathbb{E}\left[\|\nabla f(\boldsymbol{w}_k) - \nabla f(\overline{\boldsymbol{w}}_{k,\tau})\|^2\right] \\
&\quad + \eta_k \sum_{\tau=0}^{E-1} \mathbb{E}\left[\left\|\nabla f(\overline{\boldsymbol{w}}_{k,\tau}) - \sum_{q=1}^{C} \frac{1}{|\gamma_q| \times C} \sum_{i \in \gamma_q} \nabla f_i\left(\boldsymbol{w}_{k,\tau}^i\right)\right\|^2\right] \quad (29) \\
&\leq \eta_k L^2 \sum_{\tau=0}^{E-1} \mathbb{E}\left[\|\boldsymbol{w}_k - \overline{\boldsymbol{w}}_{k,\tau}\|^2\right] + \eta_k \sum_{\tau=0}^{E-1} \mathbb{E}\left[\left\|\nabla f(\overline{\boldsymbol{w}}_{k,\tau}) - \sum_{q=1}^{C} \frac{1}{|\gamma_q| \times C} \sum_{i \in \gamma_q} \nabla f_i\left(\boldsymbol{w}_{k,\tau}^i\right)\right\|^2\right] \quad (30) \\
&\leq \eta_k L^2 \sum_{\tau=0}^{E-1} \mathbb{E}\left[\|\boldsymbol{w}_k - \overline{\boldsymbol{w}}_{k,\tau}\|^2\right] \\
&\quad + \eta_k \sum_{\tau=0}^{E-1} \mathbb{E}\left[\left\|\sum_{q=1}^{C} \frac{1}{|\gamma_q| \times C} \sum_{i \in \gamma_q}\left(\nabla f_i(\overline{\boldsymbol{w}}_{k,\tau}) - \nabla f_i\left(\boldsymbol{w}_{k,\tau}^i\right)\right)\right\|^2\right] \quad (31) \\
&\leq \eta_k L^2 \sum_{\tau=0}^{E-1} \mathbb{E}\left[\|\boldsymbol{w}_k - \overline{\boldsymbol{w}}_{k,\tau}\|^2\right] + \frac{\eta_k L^2}{C} \sum_{q=1}^{C} \frac{1}{|\gamma_q|} \sum_{i \in \gamma_q} \sum_{\tau=0}^{E-1} \mathbb{E}\left[\|\overline{\boldsymbol{w}}_{k,\tau} - \boldsymbol{w}_{k,\tau}^i\|^2\right]. \quad (32)
\end{aligned}
$$

Here, eq. (29) follows using Young's inequality, eq. (30) follows from $L$-smoothness of $f$, and eq. (32) follows using Jensen's inequality. So, $A$ becomes,

$$
\begin{aligned}
A &\leq -\frac{\eta_k E}{2} \mathbb{E}\left[\|\nabla f(\boldsymbol{w}_k)\|^2\right] + \eta_k L^2 \sum_{\tau=0}^{E-1} \mathbb{E}\left[\|\boldsymbol{w}_k - \overline{\boldsymbol{w}}_{k,\tau}\|^2\right] \\
&\quad + \frac{\eta_k L^2}{C} \sum_{q=1}^{C} \frac{1}{|\gamma_q|} \sum_{i \in \gamma_q} \sum_{\tau=0}^{E-1} \mathbb{E}\left[\|\overline{\boldsymbol{w}}_{k,\tau} - \boldsymbol{w}_{k,\tau}^i\|^2\right]. \quad (33)
\end{aligned}
$$

Now using $B$ and taking an expectation over set $\gamma_{q_k}$,

$$B = -\eta_k\mu \sum_{\tau=0}^{E-1} \mathbb{E}\left[\left\langle \nabla f(\boldsymbol{w}_k), \sum_{q=1}^{C} \frac{1}{|\gamma_q| \times C} \sum_{i \in \gamma_q} \left(\boldsymbol{w}_{k,\tau}^i - \boldsymbol{w}_k\right)\right\rangle\right]$$

$$= -\frac{\eta_k\mu E}{2} \mathbb{E}\left[\|\nabla f(\boldsymbol{w}_k)\|^2\right] - \frac{\eta_k\mu}{2} \sum_{\tau=0}^{E-1} \mathbb{E}\left[\left\|\sum_{q=1}^{C} \frac{1}{|\gamma_q| \times C} \sum_{i \in \gamma_q} \left(\boldsymbol{w}_{k,\tau}^i - \boldsymbol{w}_k\right)\right\|^2\right]$$

$$+ \frac{\eta_k\mu}{2} \sum_{\tau=0}^{E-1} \mathbb{E}\left[\left\|\nabla f(\boldsymbol{w}_k) - \sum_{q=1}^{C} \frac{1}{|\gamma_q| \times C} \sum_{i \in \gamma_q} \left(\boldsymbol{w}_{k,\tau}^i - \boldsymbol{w}_k\right)\right\|^2\right] \tag{34}$$

$$= -\frac{\eta_k\mu E}{2} \mathbb{E}\left[\|\nabla f(\boldsymbol{w}_k)\|^2\right] - \frac{\eta_k\mu}{2} \sum_{\tau=0}^{E-1} \mathbb{E}\left[\|\overline{\boldsymbol{w}}_{k,\tau} - \boldsymbol{w}_k\|^2\right]$$

$$+ \frac{\eta_k\mu}{2} \sum_{\tau=0}^{E-1} \mathbb{E}\left[\|\nabla f(\boldsymbol{w}_k) - \overline{\boldsymbol{w}}_{k,\tau} + \boldsymbol{w}_k\|^2\right] \tag{35}$$

$$\leq \frac{\eta_k\mu E}{2} \mathbb{E}\left[\|\nabla f(\boldsymbol{w}_k)\|^2\right] + \frac{\eta_k\mu}{2} \sum_{\tau=0}^{E-1} \mathbb{E}\left[\|\overline{\boldsymbol{w}}_{k,\tau} - \boldsymbol{w}_k\|^2\right]. \tag{36}$$

Note eq. (34) follows due to eq. (22), while eqs. (35) and (36) follows from eq. (20) and Young's inequality respectively. Now using $M$,

$$M \leq \eta_k^2 L \mathbb{E}\left[\left\|\sum_{q=1}^{C} \frac{1}{|\gamma_{q_k}| \times C} \sum_{i \in \gamma_{q_k}} \sum_{\tau=0}^{E-1} \widetilde{\nabla} f_i\left(\boldsymbol{w}_{k,\tau}^i; \mathcal{B}_{k,\tau}^i\right)\right\|^2\right]$$

$$+ \eta_k^2 \mu^2 L \mathbb{E}\left[\left\|\sum_{q=1}^{C} \frac{1}{|\gamma_{q_k}| \times C} \sum_{i \in \gamma_{q_k}} \sum_{\tau=0}^{E-1} \left(\boldsymbol{w}_{k,\tau}^i - \boldsymbol{w}_k\right)\right\|^2\right] \tag{37}$$

$$\leq \frac{\eta_k^2 L E}{C} \sum_{q=1}^{C} \sum_{\tau=0}^{E-1} \mathbb{E}\left[\left\|\frac{1}{|\gamma_{q_k}|} \sum_{i \in \gamma_{q_k}} \widetilde{\nabla} f_i\left(\boldsymbol{w}_{k,\tau}^i; \mathcal{B}_{k,\tau}^i\right)\right\|^2\right]$$

$$+ \frac{\eta_k^2 \mu^2 L E}{C} \sum_{q=1}^{C} \sum_{\tau=0}^{E-1} \mathbb{E}\left[\left\|\frac{1}{|\gamma_{q_k}|} \sum_{i \in \gamma_{q_k}} \left(\boldsymbol{w}_{k,\tau}^i - \boldsymbol{w}_k\right)\right\|^2\right] \tag{38}$$

$$\leq \frac{\eta_k^2 L E}{C} \sum_{q=1}^{C} \frac{1}{|\gamma_q|} \sum_{i \in \gamma_q} \sum_{\tau=0}^{E-1} \mathbb{E}\left[\left\|\widetilde{\nabla} f_i\left(\boldsymbol{w}_{k,\tau}^i\right)\right\|^2\right]$$

$$+ \frac{\eta_k^2 \mu^2 L E}{C} \sum_{q=1}^{C} \frac{1}{|\gamma_q|} \sum_{i \in \gamma_q} \sum_{\tau=0}^{E-1} \mathbb{E}\left[\left\|\boldsymbol{w}_{k,\tau}^i - \boldsymbol{w}_k\right\|^2\right] \tag{39}$$

$$\leq \frac{\eta_k^2 L E}{C} \sum_{q=1}^{C} \frac{1}{|\gamma_q|} \sum_{i \in \gamma_q} \sum_{\tau=0}^{E-1} \mathbb{E}\left[\left\|\nabla f_i\left(\boldsymbol{w}_{k,\tau}^i\right)\right\|^2\right] + \eta_k^2 L E^2 \sigma^2$$

$$+ \frac{\eta_k^2 \mu^2 L E}{C} \sum_{q=1}^{C} \frac{1}{|\gamma_q|} \sum_{i \in \gamma_q} \sum_{\tau=0}^{E-1} \mathbb{E}\left[\left\|\boldsymbol{w}_{k,\tau}^i - \boldsymbol{w}_k\right\|^2\right]. \tag{40}$$

Here, eq. (37) follows from Young's inequality. Additionally, eq. (38) follows from Jensen's inequality, and Young's inequality, while eq. (39) follows from taking expectation w.r.t $\gamma_k$. Now, putting $A$, $B$, and $M$ back

in eq. (26), we get

$$\mathbb{E}\left[f(\boldsymbol{w}_{k+1})\right] \leq \mathbb{E}\left[f(\boldsymbol{w}_k)\right] - \frac{\eta_k E(1-\mu)}{2}\mathbb{E}\left[\|\nabla f(\boldsymbol{w}_k)\|^2\right] + \eta_k^2 L E^2 \sigma^2$$

$$+ \underbrace{\eta_k(L^2 + \frac{\mu}{2})\sum_{\tau=0}^{E-1}\mathbb{E}\left[\|\boldsymbol{w}_k - \overline{\boldsymbol{w}}_{k,\tau}\|^2\right]}_{(X)} + \underbrace{\frac{\eta_k L^2}{C}\sum_{q=1}^{C}\frac{1}{|\gamma_q|}\sum_{i\in\gamma_q}\sum_{\tau=0}^{E-1}\mathbb{E}\left[\|\overline{\boldsymbol{w}}_{k,\tau} - \boldsymbol{w}_{k,\tau}^i\|^2\right]}_{(Y)}$$

$$+ \underbrace{\frac{\eta_k^2 LE}{C}\sum_{q=1}^{C}\frac{1}{|\gamma_q|}\sum_{i\in\gamma_q}\sum_{\tau=0}^{E-1}\mathbb{E}\left[\|\nabla f_i\left(\boldsymbol{w}_{k,\tau}^i\right)\|^2\right]}_{(Z)} + \underbrace{\frac{\eta_k^2 \mu^2 LE}{C}\sum_{q=1}^{C}\frac{1}{|\gamma_q|}\sum_{i\in\gamma_q}\sum_{\tau=0}^{E-1}\mathbb{E}\left[\|\boldsymbol{w}_{k,\tau}^i - \boldsymbol{w}_k\|^2\right]}_{(W)}. \quad (41)$$

Using $X$ along with Lemma 2:

$$X \leq \frac{4\eta_k^3 E^2}{3C}(L^2 + \frac{\mu}{2})\sum_{q=1}^{C}\sum_{\tau=0}^{E-1}\mathbb{E}\left[\left\|\frac{1}{|\gamma_q|}\sum_{i\in\gamma_q}\nabla f_i\left(\boldsymbol{w}_{k,\tau}^i\right)\right\|^2\right] + \frac{4\eta_k^3 E^2}{3C}(L^2 + \frac{\mu}{2})\sum_{q=1}^{C}\frac{\sigma^2}{|\gamma_q|}$$

$$\leq \frac{4\eta_k^3 E^2}{3C}(L^2 + \frac{\mu}{2})\sum_{q=1}^{C}\frac{1}{|\gamma_q|}\sum_{i\in\gamma_q}\sum_{\tau=0}^{E-1}\mathbb{E}\left[\|\nabla f_i\left(\boldsymbol{w}_{k,\tau}^i\right)\|^2\right] + \frac{4\eta_k^3 E^2}{3C}(L^2 + \frac{\mu}{2})\sum_{q=1}^{C}\frac{\sigma^2}{|\gamma_q|} \quad (42)$$

$$\leq \frac{8\eta_k^3 E^3}{C}(L^2 + \frac{\mu}{2})\sum_{q=1}^{C}\frac{1}{|\gamma_q|}\sum_{i\in\gamma_q}\mathbb{E}\left[\|\nabla f_i\left(\boldsymbol{w}_k\right)\|^2\right] + \frac{4\eta_k^3 E^2}{3}(L^2 + \frac{\mu}{2})\left(\frac{2}{3} + \frac{1}{C}\sum_{q=1}^{C}\frac{1}{|\gamma_q|}\right)\sigma^2 \quad (43)$$

Equation (42) follows from Jensen's inequality and eq. (43) follows from lemma 5. Now, using $Y$ along with Lemma 3, we get

$$Y \leq \frac{8\eta_K^3 L^2 E^3}{C}\sum_{q=1}^{C}\frac{1}{|\gamma_q|}\sum_{i\in\gamma_q}\mathbb{E}\left[\|\nabla f_i\left(\boldsymbol{w}_k\right)\|^2\right] + \frac{8\eta_K^3 L^2 E^2(1+E)}{9}\sigma^2. \quad (44)$$

Using $Z$:

$$Z = \frac{\eta_k^2 LE}{C}\sum_{q=1}^{C}\frac{1}{|\gamma_q|}\sum_{i\in\gamma_q}\sum_{\tau=0}^{E-1}\mathbb{E}\left[\|\nabla f_i\left(\boldsymbol{w}_{k,\tau}^i\right)\|^2\right]$$

$$\leq \frac{6\eta_k^2 LE^2}{C}\sum_{q=1}^{C}\frac{1}{|\gamma_q|}\sum_{i\in\gamma_q}\mathbb{E}\left[\|\nabla f_i\left(\boldsymbol{w}_k\right)\|^2\right] + \frac{2\eta_k^2 LE}{3}\sigma^2. \quad (45)$$

Equation (45) follows from Lemma 5. Now using $W$,

$$W = \frac{\eta_k^2 \mu^2 LE}{C}\sum_{q=1}^{C}\frac{1}{|\gamma_q|}\sum_{i\in\gamma_q}\sum_{\tau=0}^{E-1}\mathbb{E}\left[\|\boldsymbol{w}_{k,\tau}^i - \boldsymbol{w}_k\|^2\right]$$

$$\leq \frac{4\eta_k^4 \mu^2 LE^4}{3C}\sum_{q=1}^{C}\frac{1}{|\gamma_q|}\sum_{i\in\gamma_q}\mathbb{E}\left[\|\nabla f_i\left(\boldsymbol{w}_k\right)\|^2\right] + \frac{44\eta_k^4 \mu^2 LE^3}{9}\sigma^2. \quad (46)$$

Equation (46) follows due to Lemma 4 and Lemma 5. Putting $X$, $Y$, $Z$, and $W$ back in eq. (41), we get

$$\mathbb{E}\left[f(\boldsymbol{w}_{k+1})\right] \leq \mathbb{E}\left[f(\boldsymbol{w}_k)\right] - \frac{\eta_k E(1-\mu)}{2}\mathbb{E}\left[\|\nabla f(\boldsymbol{w}_k)\|^2\right]$$

$$+ \eta_k^2 E^2\left(9\eta_k L^2 E + 4\eta_k \mu E + 6L\left(1 + \frac{4\eta_k^2 \mu^2 E^2}{18}\right)\right)\sum_{q=1}^{C}\frac{1}{|\gamma_q| \times C}\sum_{i\in\gamma_q}\mathbb{E}\left[\|\nabla f_i\left(\boldsymbol{w}_k\right)\|^2\right]$$

$$+ \eta_k^2 E \left( L \left( 1 + \frac{2\eta_k}{3} \right) + \frac{8\eta_k L^2 E (2 + E)}{9} + \frac{4\eta_k \mu E (1 + \eta_k \mu E)}{9} \right.$$

$$+ \frac{4\eta_k E}{3C} \left( L^2 + \frac{\mu}{2} \right) \sum_{q=1}^{C} \frac{1}{|\gamma_q|} \right) \sigma^2.$$

$\square$

**Lemma 2.** *For $\eta_k L E \leq \frac{1}{2}$,*

$$\sum_{\tau=0}^{E-1} \mathbb{E} \left[ \| \boldsymbol{w}_k - \overline{\boldsymbol{w}}_{k,\tau} \|^2 \right] \leq \frac{4\eta_k^2 E^2}{3C} \sum_{\tau=0}^{E-1} \sum_{q=1}^{C} \mathbb{E} \left[ \left\| \frac{1}{|\gamma_q|} \sum_{i \in \gamma_q} \nabla f_i \left( \boldsymbol{w}_{k,t}^i \right) \right\|^2 \right] + \frac{4\eta_k^2 E^2}{3C} \sum_{q=1}^{C} \frac{\sigma^2}{|\gamma_q|}$$

*Proof.*

$$\mathbb{E} \left[ \| \boldsymbol{w}_k - \overline{\boldsymbol{w}}_{k,\tau} \|^2 \right]$$

$$= \mathbb{E} \left[ \left\| \boldsymbol{w}_k - \boldsymbol{w}_k + \eta_k \sum_{q=1}^{C} \frac{1}{|\gamma_q| \times C} \sum_{i \in \gamma_q} \sum_{t=0}^{\tau-1} \widetilde{\nabla} f_i \left( \boldsymbol{w}_{k,t}^i ; \mathcal{B}_{k,t}^i \right) \right. \right.$$

$$\left. \left. + \eta_k \mu \sum_{q=1}^{C} \frac{1}{|\gamma_q| \times C} \sum_{i \in \gamma_q} \sum_{t=0}^{\tau-1} \left( \boldsymbol{w}_{k,t}^i - \boldsymbol{w}_k \right) \right\|^2 \right] \tag{47}$$

$$= \mathbb{E} \left[ \left\| \eta_k \sum_{q=1}^{C} \frac{1}{|\gamma_q| \times C} \sum_{i \in \gamma_q} \sum_{t=0}^{\tau-1} \widetilde{\nabla} f_i \left( \boldsymbol{w}_{k,t}^i ; \mathcal{B}_{k,t}^i \right) + \eta_k \mu \sum_{q=1}^{C} \frac{1}{|\gamma_q| \times C} \sum_{i \in \gamma_q} \sum_{t=0}^{\tau-1} \left( \boldsymbol{w}_{k,t}^i - \boldsymbol{w}_k \right) \right\|^2 \right]$$

$$\leq 2\eta_k^2 \mathbb{E} \left[ \left\| \sum_{q=1}^{C} \frac{1}{|\gamma_q| \times C} \sum_{i \in \gamma_q} \sum_{t=0}^{\tau-1} \widetilde{\nabla} f_i \left( \boldsymbol{w}_{k,t}^i ; \mathcal{B}_{k,t}^i \right) \right\|^2 \right] + 2\eta_k^2 \mu^2 \mathbb{E} \left[ \left\| \sum_{q=1}^{C} \frac{1}{|\gamma_q| \times C} \sum_{i \in \gamma_q} \sum_{t=0}^{\tau-1} \left( \boldsymbol{w}_{k,t}^i - \boldsymbol{w}_k \right) \right\|^2 \right]$$

$$\tag{48}$$

$$\leq \frac{2\eta_k^2}{C} \sum_{q=1}^{C} \mathbb{E} \left[ \left\| \frac{1}{|\gamma_q|} \sum_{i \in \gamma_q} \sum_{t=0}^{\tau-1} \widetilde{\nabla} f_i \left( \boldsymbol{w}_{k,t}^i ; \mathcal{B}_{k,t}^i \right) \right\|^2 \right] + 2\eta_k^2 \mu^2 \tau \sum_{t=0}^{\tau-1} \mathbb{E} \left[ \| \boldsymbol{w}_k - \overline{\boldsymbol{w}}_{k,\tau} \|^2 \right] \tag{49}$$

$$\leq \frac{2\eta_k^2 \tau}{C} \sum_{t=0}^{\tau-1} \sum_{q=1}^{C} \mathbb{E} \left[ \left\| \frac{1}{|\gamma_q|} \sum_{i \in \gamma_q} \nabla f_i \left( \boldsymbol{w}_{k,t}^i \right) \right\|^2 \right] + \frac{2\eta_k^2 \tau}{C} \sum_{q=1}^{C} \frac{\sigma^2}{|\gamma_q|} + 2\eta_k^2 \mu^2 \tau \sum_{t=0}^{\tau-1} \mathbb{E} \left[ \| \boldsymbol{w}_k - \overline{\boldsymbol{w}}_{k,t} \|^2 \right]. \tag{50}$$

Equation (47) follows from eq. (20), and eq. (48) follows from Young's inequality, eq. (49) follows from Jensen's inequality, eq. (20) and Young's inequality. Furthermore, eq. (50) follows from eq. (24). Now, summing up eq. (50) for all $\tau \in \{0, \ldots, E-1\}$, we get:

$$\sum_{\tau=0}^{E-1} \mathbb{E} \left[ \| \boldsymbol{w}_k - \overline{\boldsymbol{w}}_{k,\tau} \|^2 \right] \leq \frac{\eta_k^2 E^2}{C} \sum_{\tau=0}^{E-1} \sum_{q=1}^{C} \mathbb{E} \left[ \left\| \frac{1}{|\gamma_q|} \sum_{i \in \gamma_q} \nabla f_i \left( \boldsymbol{w}_{k,t}^i \right) \right\|^2 \right]$$

$$+ \frac{\eta_k^2 E^2}{C} \sum_{q=1}^{C} \frac{\sigma^2}{|\gamma_q|} + \eta_k^2 \mu^2 E^2 \sum_{\tau=0}^{E-1} \mathbb{E} \left[ \| \boldsymbol{w}_k - \overline{\boldsymbol{w}}_{k,\tau} \|^2 \right] \tag{51}$$

$$\sum_{\tau=0}^{E-1} \mathbb{E} \left[ \| \boldsymbol{w}_k - \overline{\boldsymbol{w}}_{k,\tau} \|^2 \right] \leq \frac{\eta_k^2 E^2}{(1 - \eta_k^2 \mu^2 E^2) C} \sum_{\tau=0}^{E-1} \sum_{q=1}^{C} \mathbb{E} \left[ \left\| \frac{1}{|\gamma_q|} \sum_{i \in \gamma_q} \nabla f_i \left( \boldsymbol{w}_{k,t}^i \right) \right\|^2 \right] + \frac{\eta_k^2 E^2}{(1 - \eta_k^2 \mu^2 E^2) C} \sum_{q=1}^{C} \frac{\sigma^2}{|\gamma_q|}$$

$$\tag{52}$$

For $\eta_k LE \leq \frac{1}{2}$ in eq. (52), we get

$$\sum_{\tau=0}^{E-1} \mathbb{E}\left[\|\boldsymbol{w}_k - \overline{\boldsymbol{w}}_{k,\tau}\|^2\right] \leq \frac{4\eta_k^2 E^2}{3C} \sum_{\tau=0}^{E-1}\sum_{q=1}^{C} \mathbb{E}\left[\left\|\frac{1}{|\gamma_q|}\sum_{i\in\gamma_q}\nabla f_i\left(\boldsymbol{w}_{k,t}^i\right)\right\|^2\right] + \frac{4\eta_k^2 E^2}{3C}\sum_{q=1}^{C}\frac{\sigma^2}{|\gamma_q|}.$$

$\square$

**Lemma 3.** *For $\eta_k \mu E \leq \frac{1}{2}$,*

$$\sum_{\tau=0}^{E-1} \mathbb{E}\left[\left\|\boldsymbol{w}_{k,\tau}^i - \overline{\boldsymbol{w}}_{k,\tau}\right\|^2\right] \leq 8\eta_k^2 E^2 \sum_{q=1}^{C}\frac{1}{|\gamma_q|\times C}\sum_{i\in\gamma_q}\sum_{\tau=0}^{E-1}\mathbb{E}\left[\|\nabla f_i(\boldsymbol{w}_k)\|^2\right] + \frac{8\eta_k^2 E^2(1+E)}{9}\sigma^2.$$

*Proof.*

$$\mathbb{E}\left[\left\|\boldsymbol{w}_{k,\tau}^i - \overline{\boldsymbol{w}}_{k,\tau}\right\|^2\right]$$

$$= \mathbb{E}\left[\left\|\left\{\boldsymbol{w}_k - \eta_k\sum_{t=0}^{\tau-1}\widetilde{\nabla} f_i\left(\boldsymbol{w}_{k,t}^i;\mathcal{B}_{k,t}^i\right) - \eta_k\mu\sum_{t=0}^{\tau-1}\left(\boldsymbol{w}_{k,t}^i - \boldsymbol{w}_k\right)\right\}\right.\right.$$

$$- \left\{\boldsymbol{w}_k - \eta_k\sum_{q=1}^{C}\frac{1}{|\gamma_q|\times C}\sum_{i\in\gamma_q}\sum_{t=0}^{\tau-1}\widetilde{\nabla} f_i\left(\boldsymbol{w}_{k,t}^i;\mathcal{B}_{k,t}^i\right)\right.$$

$$\left.\left.\left. - \eta_k\mu\sum_{q=1}^{C}\frac{1}{|\gamma_q|\times C}\sum_{i\in\gamma_q}\sum_{t=0}^{\tau-1}\left(\boldsymbol{w}_{k,t}^i - \boldsymbol{w}_k\right)\right\}\right\|^2\right] \quad (53)$$

$$= \eta_k^2 \mathbb{E}\left[\left\|\sum_{t=0}^{\tau-1}\left(\sum_{q=1}^{C}\frac{1}{|\gamma_q|\times C}\sum_{i\in\gamma_q}\widetilde{\nabla} f_i\left(\boldsymbol{w}_{k,t}^i;\mathcal{B}_{k,t}^i\right) - \widetilde{\nabla} f_i\left(\boldsymbol{w}_{k,t}^i;\mathcal{B}_{k,t}^i\right)\right)\right.\right.$$

$$\left.\left. + \mu\sum_{t=0}^{\tau-1}\left(\sum_{q=1}^{C}\frac{1}{|\gamma_q|\times C}\sum_{i\in\gamma_q}\boldsymbol{w}_{k,t}^i - \boldsymbol{w}_{k,t}^i\right)\right\|^2\right]$$

$$\leq 2\eta_k^2 \tau \sum_{t=0}^{\tau-1}\mathbb{E}\left[\left\|\sum_{q=1}^{C}\frac{1}{|\gamma_q|\times C}\sum_{i\in\gamma_q}\widetilde{\nabla} f_i\left(\boldsymbol{w}_{k,t}^i;\mathcal{B}_{k,t}^i\right) - \widetilde{\nabla} f_i\left(\boldsymbol{w}_{k,t}^i;\mathcal{B}_{k,t}^i\right))\right\|^2\right]$$

$$+ 2\eta_k^2\mu^2\tau\sum_{t=0}^{\tau-1}\mathbb{E}\left[\left\|\sum_{q=1}^{C}\frac{1}{|\gamma_q|\times C}\sum_{i\in\gamma_q}\boldsymbol{w}_{k,t}^i - \boldsymbol{w}_{k,t}^i\right\|^2\right] \quad (54)$$

$$\leq 2\eta_k^2\tau\sum_{t=0}^{\tau-1}\sum_{q=1}^{C}\frac{1}{|\gamma_q|\times C}\sum_{i\in\gamma_q}\mathbb{E}\left[\|\widetilde{\nabla} f_i\left(\boldsymbol{w}_{k,t}^i;\mathcal{B}_{k,t}^i\right)\|^2\right] + 2\eta_k^2\mu^2\tau\sum_{t=0}^{\tau-1}\left[\|\boldsymbol{w}_{k,t}^i - \overline{\boldsymbol{w}}_{k,t}\|^2\right] \quad (55)$$

$$\leq 2\eta_k^2\tau\sum_{t=0}^{\tau-1}\sum_{q=1}^{C}\frac{1}{|\gamma_q|\times C}\sum_{i\in\gamma_q}\mathbb{E}\left[\|\nabla f_i(\boldsymbol{w}_{k,t}^i)\|^2\right] + 2\eta_k^2\tau^2\sigma^2 + 2\eta_k^2\mu^2\tau\sum_{t=0}^{\tau-1}\left[\|\boldsymbol{w}_{k,t}^i - \overline{\boldsymbol{w}}_{k,t}\|^2\right]. \quad (56)$$

Here, eq. (53) follows from eqs. (19) and (20) and eq. (54) follows from Young's inequality. Now, eq. (55) follows using the fact that the second moment is greater than or equal to the variance and eq. (20). Again, eq. (56) follows from eq. (25). Then, summing up eq. (56) for all $\tau \in \{0,\ldots,E-1\}$, we get,

$$\sum_{\tau=0}^{E-1}\mathbb{E}\left[\|\boldsymbol{w}_{k,\tau}^i - \overline{\boldsymbol{w}}_{k,\tau}\|^2\right] \leq \eta_k^2 E^2 \sum_{\tau=0}^{E-1}\sum_{q=1}^{C}\frac{1}{|\gamma_q|\times C}\sum_{i\in\gamma_q}\mathbb{E}\left[\|\nabla f_i(\boldsymbol{w}_{k,\tau}^i)\|^2\right] + \frac{2\eta_k^2 E^3}{3}\sigma^2 \quad (57)$$

$$+ \eta_k^2 \mu^2 E^2 \sum_{\tau=0}^{E-1} \left[ \left\| \boldsymbol{w}_{k,\tau}^i - \overline{\boldsymbol{w}}_{k,\tau} \right\|^2 \right] \tag{58}$$

For $\eta_k \mu E \leq \frac{1}{2}$ in eq. (57), we get

$$\sum_{\tau=0}^{E-1} \mathbb{E} \left[ \left\| \boldsymbol{w}_{k,\tau}^i - \overline{\boldsymbol{w}}_{k,\tau} \right\|^2 \right] \leq \frac{4\eta_k^2 E^2}{3} \sum_{q=1}^{C} \frac{1}{|\gamma_q| \times C} \sum_{i \in \gamma_q} \sum_{\tau=0}^{E-1} \mathbb{E} \left[ \left\| \nabla f_i(\boldsymbol{w}_{k,\tau}^i) \right\|^2 \right] + \frac{8\eta_k^2 E^3}{9} \sigma^2. \tag{59}$$

Putting the results from Lemma 5 back in eq. (59), we get

$$\sum_{\tau=0}^{E-1} \mathbb{E} \left[ \left\| \boldsymbol{w}_{k,\tau}^i - \overline{\boldsymbol{w}}_{k,\tau} \right\|^2 \right] \leq 8\eta_k^2 E^3 \sum_{q=1}^{C} \frac{1}{|\gamma_q| \times C} \sum_{i \in \gamma_q} \mathbb{E} \left[ \left\| \nabla f_i(\boldsymbol{w}_k) \right\|^2 \right] + \frac{8\eta_k^2 E^2 (1+E)}{9} \sigma^2.$$

$\square$

**Lemma 4.** *For $\eta_k \mu E \leq \frac{1}{2}$,*

$$\sum_{\tau=0}^{E-1} \mathbb{E} \left[ \left\| \boldsymbol{w}_{k,\tau}^i - \boldsymbol{w}_k \right\|^2 \right] \leq \frac{4\eta_k^2 E^2}{3} \sum_{\tau=0}^{E-1} \mathbb{E} \left[ \left\| \nabla f_i \left( \boldsymbol{w}_{k,\tau}^i \right) \right\|^2 \right] + \frac{4\eta_k^2 E^2 \sigma^2}{3}.$$

*Proof.*

$$\mathbb{E} \left[ \left\| \boldsymbol{w}_{k,\tau}^i - \boldsymbol{w}_k \right\|^2 \right] = \mathbb{E} \left[ \left\| \boldsymbol{w}_k - \eta_k \sum_{t=0}^{\tau-1} \widetilde{\nabla} f_i \left( \boldsymbol{w}_{k,t}^i; \mathcal{B}_{k,t}^i \right) - \eta_k \mu \sum_{t=0}^{\tau-1} \left( \boldsymbol{w}_{k,t}^i - \boldsymbol{w}_k \right) - \boldsymbol{w}_k \right\|^2 \right] \tag{60}$$

$$= \mathbb{E} \left[ \left\| \eta_k \sum_{t=0}^{\tau-1} \widetilde{\nabla} f_i \left( \boldsymbol{w}_{k,t}^i; \mathcal{B}_{k,t}^i \right) + \eta_k \mu \sum_{t=0}^{\tau-1} \left( \boldsymbol{w}_{k,t}^i - \boldsymbol{w}_k \right) \right\|^2 \right]$$

$$\leq 2\eta_k^2 \mathbb{E} \left[ \left\| \sum_{t=0}^{\tau-1} \widetilde{\nabla} f_i \left( \boldsymbol{w}_{k,t}^i; \mathcal{B}_{k,t}^i \right) \right\|^2 \right] + 2\eta_k^2 \mu^2 \tau \sum_{t=0}^{\tau-1} \mathbb{E} \left[ \left\| \left( \boldsymbol{w}_{k,t}^i - \boldsymbol{w}_k \right) \right\|^2 \right] \tag{61}$$

$$\leq 2\eta_k^2 \tau \sum_{t=0}^{\tau-1} \mathbb{E} \left[ \left\| \nabla f_i \left( \boldsymbol{w}_{k,t}^i \right) \right\|^2 \right] + 2\eta_k^2 \tau \sigma^2 + 2\eta_k^2 \mu^2 \tau \sum_{t=0}^{\tau-1} \mathbb{E} \left[ \left\| \left( \boldsymbol{w}_{k,t}^i - \boldsymbol{w}_k \right) \right\|^2 \right]. \tag{62}$$

In eq. (60), follows due to eq. (19), and in eq. (61), follows from Young's inequality. We use eq. (25) to get eq. (62). Now, summing up eq. (62) for all $\tau \in \{0, \ldots, E-1\}$, we get,

$$\sum_{\tau=0}^{E-1} \mathbb{E} \left[ \left\| \boldsymbol{w}_{k,\tau}^i - \boldsymbol{w}_k \right\|^2 \right] \leq \eta_k^2 E^2 \sum_{\tau=0}^{E-1} \mathbb{E} \left[ \left\| \nabla f_i \left( \boldsymbol{w}_{k,\tau}^i \right) \right\|^2 \right] \eta_k^2 E^2 \sigma^2 + \eta_k^2 \mu^2 E^2 \sum_{\tau=0}^{E-1} \mathbb{E} \left[ \left\| \left( \boldsymbol{w}_{k,\tau}^i - \boldsymbol{w}_k \right) \right\|^2 \right]$$

$$\sum_{\tau=0}^{E-1} \mathbb{E} \left[ \left\| \boldsymbol{w}_{k,\tau}^i - \boldsymbol{w}_k \right\|^2 \right] \leq \frac{\eta_k^2 E^2}{(1 - \eta_k^2 \mu^2 E^2)} \sum_{\tau=0}^{E-1} \mathbb{E} \left[ \left\| \nabla f_i \left( \boldsymbol{w}_{k,\tau}^i \right) \right\|^2 \right] + \frac{\eta_k^2 E^2 \sigma^2}{(1 - \eta_k^2 \mu^2 E^2)}. \tag{63}$$

For $\eta_k \mu E \leq \frac{1}{2}$, we get

$$\sum_{\tau=0}^{E-1} \mathbb{E} \left[ \left\| \boldsymbol{w}_{k,\tau}^i - \boldsymbol{w}_k \right\|^2 \right] \leq \frac{4\eta_k^2 E^2}{3} \sum_{\tau=0}^{E-1} \mathbb{E} \left[ \left\| \nabla f_i \left( \boldsymbol{w}_{k,\tau}^i \right) \right\|^2 \right] + \frac{4\eta_k^2 E^2}{3} \sigma^2.$$

$\square$

**Lemma 5.** *For $\eta_k L E \leq \frac{1}{2}$,*

$$\sum_{\tau=0}^{E-1} \mathbb{E} \left[ \left\| \nabla f_i \left( \boldsymbol{w}_{k,\tau}^i \right) \right\|^2 \right] = 6 \sum_{\tau=0}^{E-1} \mathbb{E} \left[ \left\| \nabla f_i (\boldsymbol{w}_k) \right\|^2 \right] + \frac{2}{3} \sigma^2.$$

*Proof.*

$$\sum_{\tau=0}^{E-1} \mathbb{E}\left[\left\|\nabla f_i\left(\boldsymbol{w}_{k,\tau}^i\right)\right\|^2\right] = \sum_{\tau=0}^{E-1} \mathbb{E}\left[\left\|\nabla f_i(\boldsymbol{w}_{k,\tau}^i) - \nabla f_i(\boldsymbol{w}_k) + \nabla f_i(\boldsymbol{w}_k)\right\|^2\right]$$

$$\sum_{\tau=0}^{E-1} \mathbb{E}\left[\left\|\nabla f_i\left(\boldsymbol{w}_{k,\tau}^i\right)\right\|^2\right] \leq 2E\mathbb{E}\left[\left\|\nabla f_i(\boldsymbol{w}_k)\right\|^2\right] + 2\sum_{\tau=0}^{E-1} \mathbb{E}\left[\left\|\nabla f_i(\boldsymbol{w}_{k,\tau}^i) - \nabla f_i(\boldsymbol{w}_k)\right\|^2\right] \tag{64}$$

$$\sum_{\tau=0}^{E-1} \mathbb{E}\left[\left\|\nabla f_i\left(\boldsymbol{w}_{k,\tau}^i\right)\right\|^2\right] \leq 2E\mathbb{E}\left[\left\|\nabla f_i(\boldsymbol{w}_k)\right\|^2\right] + 2L^2\sum_{\tau=0}^{E-1} \mathbb{E}\left[\left\|\boldsymbol{w}_{k,\tau}^i - \boldsymbol{w}_k\right\|^2\right]. \tag{65}$$

Equation (64) follows from Young's inequality and eq. (65) follows from $L$-smoothness of $f_i$. Now, putting the results of Lemma 4 back in eq. (65), we get

$$\sum_{\tau=0}^{E-1} \mathbb{E}\left[\left\|\nabla f_i\left(\boldsymbol{w}_{k,\tau}^i\right)\right\|^2\right] \leq 2E\mathbb{E}\left[\left\|\nabla f_i(\boldsymbol{w}_k)\right\|^2\right] + \frac{8\eta_k^2 L^2 E^2}{3}\sum_{\tau=0}^{E-1} \mathbb{E}\left[\left\|\nabla f_i\left(\boldsymbol{w}_{k,\tau}^i\right)\right\|^2\right] + \frac{8\eta_k^2 L^2 E^2}{3}\sigma^2. \tag{66}$$

For $\eta_k L E \leq \frac{1}{2}$, we get

$$\sum_{\tau=0}^{E-1} \mathbb{E}\left[\left\|\nabla f_i\left(\boldsymbol{w}_{k,\tau}^i\right)\right\|^2\right] \leq 6E\mathbb{E}\left[\left\|\nabla f_i(\boldsymbol{w}_k)\right\|^2\right] + \frac{2}{3}\sigma^2. \tag{67}$$

$\square$

