# OpenReview forum: "Equitable Federated Learning with Activation Clustering"
_TMLR — Rejected by TMLR_

### Review · Reviewer_Nqiu · 2024-12-20

**Summary Of Contributions:**

This paper proposes clustering clients based on the similarity matrix derived from the pre-classifier features of DNNs to leverage the inherent clustering properties in FL. It then ensures that each cluster, obtained through K-means clustering applied to this similarity matrix, receives equal importance during client aggregation.

**Audience:**

Yes

**Claims And Evidence:**

No

**Requested Changes:**

See above

**Strengths And Weaknesses:**

Strengths:
- The paper is very easy to read.
- The experiments were conducted on several datasets and neural network architectures.

Weaknesses:
- The motivation is not very clear to me. While I agree with the authors that clients in federated learning (FL) can naturally exhibit different clustering behaviors, making it reasonable to cluster clients and learn a cluster-specific server model [4], this paper only uses the clustering results to derive a new regularizer for weighing the clients. It does not produce a cluster-level server model. I am curious about the reasoning behind this decision. Will the authors consider learning a cluster-specific server model?

- The method section feels too brief and could benefit from more detailed explanations.

- In Algorithm 1, please specify that $a_{k, E}^i$​ refers to the activation vector before line 11. Additionally, I find line 20 unclear; the right-hand side of the equation seems unrelated to $i$. Could you clarify this? Also, does line 20 imply that the sum of $p_k^i$​ is not equal to 1?

- The so-called activation vector (the features from the last layer before the classification layer) is effective at capturing client-specific characteristics, particularly in heterogeneous settings, has been extensively studied, e.g., [1] and [2].

- How was C (the number of clusters) decided in Figure 2? Why do MNIST and CIFAR-10 use the same number of clusters, given that CIFAR-10 is more complex than MNIST?

- I understand that during the data preparation process, the authors divide the data into C clusters. However, I am curious about the number of clusters used in the K-means algorithm. Was this parameter tuned, and if so, how?

- It is well-documented that the regularization parameter in FedProx can be challenging to tune and may underperform compared to other FL algorithms [3], such as SCAFFOLD and FedDyn. Have the authors considered testing other types of FL algorithms?

- Although the authors experimented with multiple datasets, the number of clients is limited to only 10 for the CIFAR-10/100 datasets. Moreover, the data preparation process appears overly rigid. For instance, dividing the data into 2 clusters where the first 4 clients receive data from the first four classes and the next 6 clients receive data from the following six classes seems hardcoded. What is the rationale behind this approach? Have the authors tried randomly sampling classes for different clients instead?

- The experimental design seems somewhat ad hoc. Could the authors propose a real-world scenario where this experimental setup would be applicable?

[1] Partial variance reduction improves non-convex federated learning on heterogeneous data. CoRR, abs/2212.02191, 2022a

[2] Classifier calibration for federated learning with non-iid data. In Neural Information Processing Systems, 2021.

[3] SCAFFOLD: Stochastic controlled averaging for federated learning.Proceedings of the 37th International Conference on Machine Learning

[4] An Efficient Framework for Clustered Federated Learning. In Neural Information Processing Systems,  2020

---

> ### Author Response · Authors · 2025-03-13
> **Replies to Reviewer Nqiu: Part 1**
>
> First, we would like to express our gratitude to the reviewer for their insightful feedback. We have made an effort to respond to your inquiries and hope we have covered everything.
>
> >**Weakness:**
>
> - *The motivation is not very clear to me. While I agree...* :
>
> Thank you for your insightful question. In this work, our goal is to develop a single global model that performs equitably across all clients. To achieve this, we utilize activation vectors as side information to cluster clients and assign equitable weights accordingly. As the reviewer correctly pointed out, we do not create personalized models for each cluster. The key motivation behind this choice is to ensure that all clients receive a uniform model quality, avoiding disparities in performance. Additionally, maintaining a single global model is computationally more efficient compared to managing multiple personalized models. That said, developing personalized models that address these challenges is an exciting direction for future work, and we plan to explore this in subsequent research.
>
> - *Will the authors consider learning a cluster-specific server model?*
>
> We appreciate the reviewer’s suggestion. We have conducted preliminary experiments on the CIFAR-10 dataset in the full participation setting, where data distribution is clustered among 10 clients, as described in Section 5.1. The algorithmic framework remains unchanged, except we now train cluster-specific models, where clients within each cluster receive the corresponding model. We provide the results for this approach in the tabulated section below.
>
> **Table: Comparison of IFCA vs. Equitable-FL (personalized).**
> The table shows the average test accuracy and \($\sigma_{Acc}$\) of 3 independent runs over 100 global communication rounds. The results are produced using the ResNet-18 model architecture.  Bold numbers indicate the best results.
>
> | Method                         | CIFAR-10 Acc (\%)        | CIFAR-10 \($\sigma_{Acc}$  ($\downarrow)$\) |
> |--------------------------------|-------------------------|--------------------------------|
> | **IFCA**                       | \(80.30 \pm 8.65\)      | \(8.45 \pm 8.11\)            |
> | **Equitable-FL (personalized)** | **\(87.33 \pm 0.20\)**  | **\(2.43 \pm 0.06\)**        |
>
> - *The method section feels too brief and could benefit from more detailed explanations.:*
>
>  Thank you for raising this point. We have addressed this concern in the revised version by providing a detailed explanation of the Similarity and getprobs subroutines. Additionally, we have included the precise definition of fairness in Section 4.2. Please find the changes in the General answer.
>
> - *In Algorithm 1, please specify that $a^i_{k,E}$ refers to the activation vector before line 11.*
>
> Thank you for the feedback; we have included it in the revised version of the paper, in Section 1, third paragraph. Please find the details below in quotes.
>
> "It proposes a clustering paradigm based on *activation vectors* ($a_{k,E}^i$, See Definition 1) that promote group fairness"
>
> - *Additionally, I find line 20 unclear; ...:*
>
> The right-hand side of equation 20 is cluster-dependent rather than client-specific because we assign equal weights to all clients within the same cluster. Specifically, $p^i_k$ is determined by the total number of clients in the cluster rather than individual client characteristics. Please refer to Definition 3 in the rebuttal above. This ensures that each cluster contributes equally to the global update, aligning with our fairness objective. So, it ensures that the sum of probabilities across all clients remains equal to 1. We have changed that as well. Please find it in the General answer.
>
> - *The so-called activation vector ...*
>
> We appreciate the reviewer’s insightful comment regarding the use of activation vectors in federated learning under heterogeneous settings. Prior works, such as [1,2], have analyzed how deep representations are affected by non-IID client distributions and demonstrated that activation vectors retain useful structure even in heterogeneous settings. Our approach builds upon these findings but differs in key ways. While [2] propose classifier calibration (CCVR) to mitigate classifier bias and [1] suggest partial variance reduction using SCAFFOLD(FedPVR) for stabilizing deeper layers, our work leverages activation vectors for clustering clients. By computing similarity-based groupings, we ensure that clients with similar learned representations contribute equitably to the model update. Furthermore, activation vectors serve as a compact yet expressive representation of client characteristics, making them well-suited for our clustering-based reweighting approach. Our experimental results confirm that this leads to more stable and fairer learning dynamics compared to conventional FL baselines.
>
> We have incorporated the related works section into the revised manuscript. The details are in the quotes in the next part.

---

> ### Author Response · Authors · 2025-03-13
> **Replies to Reviewer Nqiu: Part 2**
>
> "Additionally, prior works such as [1,2] analyze how deep representations are affected by non-IID client distributions. [2] propose classifier calibration (CCVR) to mitigate classifier bias, and [1] suggest partial variance reduction using SCAFFOLD (FedPVR) for stabilizing deeper layers.''
>
> - *How was C (the number of clusters) decided...*:
>
> In our paper, $C$ is a hyperparameter that requires tuning. Specifically, we tested different values of $C$ and selected the one that provided a good trade-off between fairness and accuracy. Additionally, Figure 2 presents an NMI comparison plot, which we generated following the experimental design outlined in Section 5.  We used the same number of clusters for both MNIST and CIFAR-10 as this choice allowed us to isolate the effect of clustering itself without introducing dataset-specific variations. To further demonstrate the flexibility of our framework, we also tested CIFAR-100 with 3 clusters and FEMNIST with 5 clusters. These experiments illustrate that our approach generalizes well across datasets with different levels of complexity. The NMI comparison plot in Figure 2 evaluates clustering quality across these settings, reinforcing the robustness of our method.
>
> - *I understand that during the data preparation...*:
>
> Thank you for your question. In our approach, the number of clusters $C$ used in the K-means algorithm is a hyperparameter that requires tuning. As mentioned in Remark 3, we select $C$ based on empirical validation to ensure meaningful client groupings.  We evaluate different values of $C$ and choose the one that provides the best trade-off between the fairness and accuracy we achieve.
>
>
> - *It is well-documented that the regularization parameter in FedProx ...*:
>
> Thank you for your insightful question. We acknowledge that the regularization parameter in FedProx can be challenging to tune and that alternative federated learning algorithms, such as SCAFFOLD and FedDyn, have demonstrated strong performance in addressing client drift. However, FedProx provides a simple yet effective mechanism for handling statistical heterogeneity while maintaining a single global model. Since our objective is to improve fairness across clients, FedProx serves as a natural baseline for assessing the impact of our methodology. Additionally, methods like SCAFFOLD and FedDyn require additional communication overhead and storage for control variates or auxiliary variables, which could introduce extra complexity. Given our focus on clustering and equitable weighting, we prioritized maintaining a lightweight framework that can be easily adapted to different FL settings.
> Nevertheless, we experimented by applying our clustering and client-weighting framework to SCAFFOLD on the CIFAR-10 dataset. Before presenting the results, we outline the experimental setup. We utilized a standard non-IID partitioning scheme, where the dataset was distributed among 100 clients, each receiving 500 samples containing data from at most two labels. The experiment was performed using the ResNet-18 model over 200 communication rounds, with 20\% of the clients participating in each round. Please refer to the table below for the results. We observe that our method reduces the deviation in accuracy across clients, further validating its effectiveness beyond FedProx.
>
> **Table: Comparison of SCAFFOLD vs. Equitable-FL (SCAFFOLD).**
> The table shows the average test accuracy and \($\sigma_{Acc}$\) of 3 independent runs over 200 global communication rounds. The results are produced using the ResNet-18 model architecture.
> Bold numbers indicate the best results.
>
> | Method                     | CIFAR-10 Acc (\%)         | CIFAR-10 $\sigma_{Acc}$ ($\downarrow)$ |
> |----------------------------|-------------------------|--------------------------------|
> | **SCAFFOLD**               | **\(38.64 \pm 0.90\)**  | \(17.19 \pm 1.35\)            |
> | **Equitable-FL (SCAFFOLD)** | \(37.31 \pm 3.00\)      | **\(16.64 \pm 1.08\)** |
>
>
> - *Although the authors experimented with multiple datasets...*:
>
> Thank you for your thoughtful question. Our choice of using 10 clients for the CIFAR-10/100 datasets was primarily motivated by computational feasibility while ensuring meaningful heterogeneity in client data distributions. While we acknowledge that real-world federated learning systems may involve larger numbers of clients, our setup effectively captures the key challenges of non-IID data distribution, which is central to our study. Our goal was to simulate a realistic heterogeneous data setting where clients receive disjoint and imbalanced subsets of data. By allocating data in a structured manner (e.g., 4 clients receiving data from 4 classes and 6 clients from the remaining 6 classes), we ensure a clear distinction between clusters. This structured partitioning enables us to evaluate whether our clustering method effectively groups similar clients clearly.

---

> ### Author Response · Authors · 2025-03-13
> **Replies to Reviewer Nqiu: Part 3**
>
> Additionally, we experimented by applying our clustering and client-weighting framework with Fedprox to the CIFAR-10 dataset. Before presenting the results, we outline the experimental setup. We utilized a standard non-IID partitioning scheme, where the dataset was distributed among 100 clients, each receiving 500 samples containing data from at most two labels. The experiment was performed using the ResNet-18 model over 200 communication rounds, with 20\% of the clients participating in each round. Please refer to the tabulated results below. Also, please refer to the Table in General answer for experiments with CelebA dataset.
>
> **Table: Performance Comparison for non-IID Data Distribution.**
> We compare the accuracy of different algorithms against **Equitable-FL**. The table presents the average test accuracy and standard deviation ($\sigma_{Acc}$) over three independent runs for 200 global communication rounds using ResNet-18. Bold numbers indicate the best results. Additionally, results in the centralized setting are included for reference but are not highlighted as best results.
>
> | Method                     | CIFAR-10 Acc (\%)   | CIFAR-10 $\sigma_{Acc}$ $(\downarrow$) |
> |----------------------------|-------------------------|--------------------------------|
> | **Centralized**            | \(85.67 \pm 0.56\)      | \(5.32 \pm 0.22\)  |
> | **FedProx**                | \(51.78 \pm 1.73\)      | \(19.59 \pm 1.05\) |
> | **Cluster2**               | **\(53.81 \pm 0.83\)**  | **\(18.46 \pm 0.62\)** |
> | **Pow-d**                  | \(51.16 \pm 1.54\)      | \(19.62 \pm 0.95\) |
> | **FairFed**                | \(39.68 \pm 0.59\)      | \(25.01 \pm 0.69\) |
> | **GIFAIR**                 | \(49.75 \pm 1.67\)      | \(20.64 \pm 1.13\)            |
> | **Equitable-FL (FedProx)** | \(52.68 \pm 1.18\)      | \(18.98 \pm 1.08\)            |
>
>
> - *The experimental design seems somewhat ad hoc. Could the authors propose a real-world scenario where this experimental setup would be applicable?*
>
> Thank you for your question. Our experimental setup is designed to mimic real-world federated learning scenarios where clients exhibit heterogeneous data distributions. In a federated network, devices such as smartphones, IoT sensors, and even Low Earth Orbit (LEO) satellites collect data under diverse conditions.
>
> A key example is LEO satellite constellations, where different satellites capture imagery, weather patterns, or communication signals from distinct geographical regions. Due to environmental variations (e.g., cloud coverage, terrain differences, atmospheric interference), satellite data distribution can be highly non-IID. Clustering LEO satellites based on shared characteristics, such as orbital regions or sensor types, enables more efficient and equitable model training, ensuring no satellite data dominates the learning process while reducing communication overhead in constrained network environments.
>
> Similarly, this setup extends to terrestrial applications, such as clustering IoT devices or mobile users based on geographical regions and hardware similarities, facilitating personalized yet fair model updates in real-world federated learning deployments.

---

### Review · Reviewer_N8c9 · 2025-01-01

**Summary Of Contributions:**

The clients consider a federated learning setup, where each client $i\in [n]$ has a loss function $f_i:\mathbb{R}^d \to\mathbb{R}$.
They assume that clients belong to $C$ clusters, and the objective is to minimize,
$$
\begin{align*}
f(w) \coloneqq \frac{1}{C}\sum_{q=1}^C \frac{1}{\lvert\gamma_q\rvert}\sum_{i\in \gamma_q} f_i(w)
\end{align*}
$$
where $\gamma_q$ is the subset of clients in $q^{th}$ cluster. The main contributions of this paper are listed below.

The objective ensures group fairness, in the sense that every cluster gets equal weightage.

- The clusters $\{\gamma_q\}_{q\in [C]}$ are unknown and the authors propose Algorithm 1, Equitable-FL, to find the clusters via clustering the last-layer activations.
In each round of their algorithm, they run FedProx updates (Li et al 2020) locally on each client for $E$ steps, then find the activation-based clustering to assign weights to each client's updated model during server aggregation.

- They analyze convergence rates assuming the **correct clustering is recovered at every step**, under smoothnes, non-negativity and bounded noise variance assumptions for full client participation in Theorem 1. They obtain $\mathcal{O}(\frac{1}{\sqrt{K}})$ convergence rate for non-convex objectives.


- Further, they test their algorithm against centralized, federated and fair federated baselines on clustered MNIST, CIFAR-10, CIFAR-100 and FEMNIST datasets.
Their algorithm outperforms most baselines in terms of test accuracy of global model, and outperforms all baselines in terms of the standard deviation of the clients' test accuracy.

**Audience:**

Yes

**Broader Impact Concerns:**

None.

**Claims And Evidence:**

No

**Requested Changes:**

Please fix the weaknesses. In terms of importance to my recommendation, I feel the theoretical analysis at this stage makes extremely strong assumptions, without explicitly stating them. With these assumptions, the theoretical analysis is not for Equitable-FL, rather for FedProx on any clustering of the clients. Therefore, the theory is too weak.

**Strengths And Weaknesses:**

### Strengths

- **Clustering for fairness**: The idea of using clustering based on model activations and subsequently weighing each cluster equally in the global update is a nice method to ensure group fairness among different sets of clients.


- **Experiments**: The experiments are exhaustive including both simulated and real federated datasets. Further, their method outperforms all baselines except centralized in terms of standard deviation of client accuracy, which is the appropriate fairness metric, while achieving close to best test accuracy on all clients.


### Weaknesses
- **Presentation**: The authors do not define the notion of "fairness" which they want to achieve formally in their problem setup. Do they want to minimize the standard deviation of all clients or just among clients in different clusters? Based on their algorithm, it seems that they want to minimize the deviation between clients in different clusters, so they weigh them equally. If this is not the case and they want to minimize the deviation of all clients, then why is using a clustered model justified, as clients in the same cluster might have different accuracies. Further, defining the precise notion of "fairness" in this setting, should allow the authors to argue why baseline methods should not work. Additionally, the authors mention the "Similarity" and "getprobs" subroutines in the algorithm, but do not provide their formal definitions. Additionally, the authors have defined a client disagreement metric $CD_{k+1}$ in Eq (9) but they have not reported its value in any case.
    1. **Typos**: $\eta_k \mu E \leq \frac{1}{2}$ in Lemma 5, $\nabla f_i(w_{k,\tau}^i)$ in Lemma 2, $\zeta_1$ in Eq (13) should be $\zeta$.


- **Algorithm**: It is unclear why the authors need to perform FedProx updates, with the additional regularization term $\mu$, as opposed to the more common FedAvg update with $\mu=0$. In the FedProx paper (Li et al 2020), the motivation for FedProx updates is to mitigate heterogeneity among clients by solving a regularized problem. Over here, the clustering procedure does not necessitate FedProx updates and should account for heterogeneity, so I do not understand why FedProx updates are explicitly needed. Further, Remark 1 is not always correct, as for larger models, one might prefer training only the last layers in a federated manner and using a pre-trained model for the initial layers.


- **Theory**:
    1. **Every round should obtain the same and correct clustering**: The authors should clarify whether $f$ in Theorem 1 refers to the original formulation (1) or the clustered formulation (7). In Theorem 1, the authors assume that all clients participate, as the update step, see Eq (20), involves all clients. Then, the authors further assume that in each step, there is no misclustering error, as the contribution of each client is in its appropriate cluster. This is a very strong requirement as it implies that even in the first round, after just a few local 4steps, the activation vector-based clustering recovers the complete correct clustering. Further, none of the Assumptions actually mention this, and Assumption 4 which requires conditions on partial participation isn't even used. Therefore, Theorem 1 is the convergence rate of FedProx algorithm with reweighted updates according to some fixed clustering, not that of Equitable-FL, where the clustering needs to be computed.
    2. **Lack of Heterogeneity assumption:** The paper has no heterogeneity assumption, while the technique they follow from (Das et al 2022) requires one, see Assumption 4 therein. Why don't the authors require any heterogeneity assumption. From my understanding, it looks like using regularization via FedProx updates allows them to remove this requirement. Further, the term $f(w_0)$ in Theorem 1 can be extremely large, as we can add a large enough constant, possibly proportional to $K$ to all $f_i$. This does not change the minima or the gradient of any $f_i$, but it would make the rates in Theorem 1 vacuous. For existing proofs in literature for instance (Karimireddy et al 2020) which obtain $f(w_0) - f(w^\star)$ instead of just $f(w_0)$, this won't be a problem as the additional term would appear in both $f(w_0)$ and $f(w^\star)$ and would thus cancel out. Ideally for a clustered setup, the clients in same cluster should have low heterogeneity and those in different clusters should have high heterogeneity. As there are no such assumptions, the convergence rates should hold for any arbitrary clustering over the clients, even those clustering where heterogeneous clients are placed in the same cluster.

- **Literature survey**: From the previous point, the authors require full client participation, therefore, (Chen & Vikalo 2024) becomes a valid baseline, as they also use all clients at every round. Additionally, as the theoretical results also require knowledge of correct clustering after just $1$ round, the methods (Yue et al 2023) that require the knowledge of clustering do not seem as bad baselines anymore.

---

> ### Author Response · Authors · 2025-03-13
> **Replies to Reviewer N8c9: Part 1**
>
> Firstly, we want to thank the reviewer for their valuable feedback. We have tried to answer your questions and hopefully have addressed them all.
>
> >**Weakness**
>
> **Presentation:** Please refer to the General answers, where we addressed the questions about fairness and the algorithms. In the paper, we presented the plots for client disagreement in Figure 3 and included them in the Appendix. Additionally, we would like to thank the reviewers for pointing out the typos.
>
> **Algorithm:** Our approach to equitable client weighting provides clustering side information that can be seamlessly integrated into any framework, and we specifically chose FedProx for this purpose. While our strategy ensures equal weighting across clusters, FedProx further minimizes intra-client deviations within each cluster, contributing to a fairer global model.
>
> We also appreciate the question regarding the validity of Remark 1. As the reviewer rightly pointed out, one might prefer to train only the final layers and transmit the relevant information to the server for larger models. Even in such cases, the activation vectors remain significantly smaller than other representation vectors, such as gradients. We acknowledge the reviewer’s comment and will ensure these clarifications are incorporated in the revised version. Please find the modified changes in the double quotes below.
>
> "**Remark 1**: Sending the additional $a^i_{k,E}$ to the server does not pose a severe communication cost in comparison to the model parameters because they are mostly of size $\mathcal{O}(10^2)$ in practice, which is considerably smaller than typical model sizes. Furthermore, for larger models where only the final layers are trained in a federated manner, using a pre-trained backbone, the activation vectors remain significantly smaller than gradients or other full-layer representation vectors. This makes them a computationally and communication-efficient choice for clustering and similarity-based weighting."
>
> **Theory**
> - **Every round should obtain the same and correct clustering**: Our theoretical analysis is based on the formulation presented in Equation (7). We conduct our analysis such that it accounts for partial client participation. We understand the potential confusion regarding Equation (20),
>
> $$
> \overline{\bm{w}}_{k,\tau} = \bm{w}_{k} -\eta_k\sum_{q=1}^{C} \frac{1}{|\gamma_{q}| \times C}\sum_{i\in \gamma_{q}} \sum_{t=0}^{\tau-1}\widetilde{\nabla} f_i\left(\bm{w}_{k,t}^{i};\mathcal{B}^{i}_{k, t}\right) - \eta_k \mu \sum_{q=1}^{C} \frac{1}{|\gamma_{q}| \times C}\sum_{i\in \gamma_{q}} \sum_{t=0}^{\tau-1}\left(\bm{w}_{k,t}^{i}-\bm{w}_k\right)
> $$
>
> ,but we are just defining terms to be used later in the analysis. In Equation (21),
>
> $$ \bm{w}_{k+1} = \bm{w}_{k} -\eta_k\sum_{q=1}^{C} \frac{1}{|\gamma_{q_k}| \times C}\sum_{i\in \gamma_{q_k}} \sum_{\tau=0}^{E-1}\widetilde \nabla f_i\left(\bm{w}_{k,\tau}^{i};\mathcal{B}^{i}_{k, \tau}\right)
>         -\eta_k \mu \sum_{q=1}^{C} \frac{1}{|\gamma_{q_k}| \times C}\sum_{i\in \gamma_{q_k}} \sum_{\tau=0}^{E-1}\left(\bm{w}_{k,\tau}^{i}-\bm{w}_k\right)$$
>
> ,we explicitly provide the global update while incorporating partial participation, denoted by $\gamma_{{q}_k}$, representing the number of clients in each cluster $q$ at round $k$.
>
> **Then the authors further assume...*: We acknowledge the reviewer’s concern regarding the assumption of no misclustering error. This assumption is made strictly for theoretical analysis, but we do not impose any such constraint in our experiments. We will ensure that this clarification is explicitly stated in Assumption 4 in the revised version. Please find the modified changes to Assumption 4 in quotes below.
>
> "**Assumption 4 (Misclustering and Existence of cluster)}:** Assuming a system comprising $C$ clusters to which all $n$ clients are allocated, this assumption aligns with the inherent system partitions, for instance, clients segmented by diverse demographic regions. Each cluster, denoted by $\gamma_q$, encapsulates a subset of participants, with $q$ signifying the specific cluster. We assume that a perfect clustering scenario exists for theoretical analysis, meaning that each client is always assigned to the correct cluster with no misclustering error. In practice, our algorithm constructs the clusters at each round, and misclustering can occur, especially in early training rounds. Further, in each $k-th$ communication round, a subset of  $r$ clients are selected and assigned to $C$ clusters, ensuring that at least one client per cluster is present, though this constraint is relaxed during experiments."

---

> ### Author Response · Authors · 2025-03-13
> **Replies to Reviewer N8c9: Part 2**
>
> - **Lack of Heterogeneity assumption:**
>
> **The paper has no heterogeneity assumption...:*  We appreciate the reviewer’s detailed analysis and the insightful questions raised. We understand the concern regarding Assumption 4. However, we do not need to make this assumption because, during our analysis, we can avoid it by using Jensen's inequality and smoothness of $f_i$'s. Lets start by reiterating the equation of Assumption 4 in (Das et al 2022) but with our objective, i.e., Equation (7),
>
> $$\mathbb{E}\left[\left\|\nabla f(\overline{\bm{w}}_{k,\tau}) - \sum_{q=1}^{C} \frac{1}{|\gamma_{q}| \times C}\sum_{i\in \gamma_{q}}\nabla f_i\left(\bm{w}_{k,\tau}^{i}\right)\right\|^2\right] = \mathbb{E}\left[\left\|\sum_{q=1}^{C} \frac{1}{|\gamma_{q}| \times C}\sum_{i\in \gamma_{q}}\left(\nabla f_i(\overline{\bm{w}}_{k,\tau}) - \nabla f_i\left(\bm{w}_{k,\tau}^{i}\right)\right)\right\|^2\right].$$
>
> Now by applying Jensen's inequality and using smoothness of $f_i$ we have
>
> $$\mathbb{E}\left[\left\|\nabla f(\overline{\bm{w}}_{k,\tau}) - \sum_{q=1}^{C} \frac{1}{|\gamma_{q}| \times C}\sum_{i\in \gamma_{q}}\nabla f_i\left(\bm{w}_{k,\tau}^{i}\right)\right\|^2\right] \leq \frac{L^2}{C} \sum_{q=1}^{C} \frac{1}{|\gamma_{q}|}\sum_{i\in \gamma_{q}}\mathbb{E}\left[\left\|\overline{\bm{w}}_{k,\tau} - \bm{w}_{k,\tau}^{i}\right\|^2\right].$$
>
> Thus we do not need to use Assumption 4 from Das et al. 2022.
>
> **Further, the term $f(w_0)$ in Theorem 1 can be extremely large...:*
>
>  We appreciate the reviewer’s insightful concern about the potential arbitrariness of $f(w_0)$ in Theorem 1. We conducted our analysis based on the work of Das et al.(2022), where by utilizing the non-negativity of $f_i$'s, we can drop the $f_i^*$ term from the analysis. Our analysis's core quantity of interest is $\|| \nabla f(w)\||^2$, so adding a constant to these functions does not affect the gradient or the optimization trajectory. Thus, even if an artificial shift were introduced, it would not change our bounds' optimization behavior or applicability.
>
> **Ideally for a clustered setup, the clients in same cluster...:*
>
> We acknowledge the reviewer's concern regarding heterogeneity within clusters. Our theoretical analysis assumes a perfect clustering scenario (as described in the modified Assumption 4), which simplifies the derivation but does not impose this restriction in practical implementations. In reality, misclustering may occur, meaning some heterogeneous clients may be placed in the same cluster. However, this does not invalidate our convergence results, as they are designed to hold in the presence of arbitrary clustering.
>
> Furthermore, the degradation in performance due to the misclustering is not unique to our approach. Our clustering-based approach does not exacerbate this issue; instead, it aims to mitigate it by ensuring that clients with similar data distributions receive similar updates. While the theoretical analysis assumes ideal clustering, our experiments demonstrate that clustering-based aggregation consistently improves performance over other baselines (see Section 5.3). Additionally, clustering improves fairness across clients by reducing disparities in model accuracy (see Section 5.4), which is a key motivation for our approach. These results confirm that even though convergence holds for arbitrary clustering, our method leads to meaningful performance improvements over naive FL methods in practice.
>
> - Literature Survey: Thank you for your feedback. As mentioned in the Related Work section, the approach in Chen \& Vikalo (2024) is an orthogonal direction focusing on client sampling. In contrast, our framework does not require full client participation, and all our experimental results are conducted under partial client participation settings. Regarding Yue et. al. 2023, we have explicitly compared our results with theirs, as presented in the Experiments section. Additionally, we present the results here to facilitate a comprehensive comparison (please refer to part 3 for it).

---

> ### Author Response · Authors · 2025-03-13
> **Replies to Reviewer N8c9: Part 3**
>
> **Table: Comparison GIFAIR vs. Equitable-FL.**
> The report shows the average test accuracy and \($\sigma_{Acc}$\) of 3 independent runs over 100 global communication rounds. The results are produced using the ResNet-18 model architecture. Bold numbers indicate the best results. Additionally, results in the centralized setting row are just for reference; hence, we have not indicated them as the best results.
>
> | Method          | MNIST Acc (\%) | MNIST \($\sigma_{Acc}$\) ($\downarrow$) | CIFAR-10 Acc (\%) | CIFAR-10 \($\sigma_{Acc}$\) ($\downarrow$)  | CIFAR-100 Acc (\%) | CIFAR-100 \($\sigma_{Acc}$\) ($\downarrow$)  | FEMNIST Acc (\%) | FEMNIST \($\sigma_{Acc}$\) ($\downarrow$)  |
> |----------------|--------------|--------------------------------|----------------|--------------------------------|----------------|--------------------------------|----------------|--------------------------------|
> | **Centralized**  | \(99.43 \pm 0.02\) | \(0.10 \pm 0.02\)  | \(83.60 \pm 0.10\) | \(5.41 \pm 0.21\)  | \(65.90 \pm 0.21\) | \(0.10 \pm 0.16\)  | \(90.51 \pm 0.07\) | \(11.67 \pm 0.01\) |
> | **GIFAIR**       | \(96.02 \pm 0.33\) | \(2.86 \pm 0.31\)  | \(64.41 \pm 0.56\) | \(20.25 \pm 1.21\) | \(55.39 \pm 0.14\) | \(17.08 \pm 0.13\) | **\(69.55 \pm 0.62\)** | \(25.53 \pm 0.61\) |
> | **Equitable-FL** | **\(96.95 \pm 0.40\)** | **\(1.65 \pm 0.53\)**  | **\(69.40 \pm 0.48\)** | **\(7.83 \pm 0.71\)**  | **\(55.65 \pm 0.15\)** | **\(13.67 \pm 0.39\)**  | \(67.02 \pm 0.42\) | **\(24.11 \pm 0.33\)** |

---

### Review · Reviewer_d8r7 · 2025-03-03

**Summary Of Contributions:**

The paper proposes a new FL clustering method based on the outputs of the second-to-last layer of the model on the clients' data. Given these values, the algorithm groups the clients and show that performance and the agreement (standard deviation) between client model performances improves compared to 5 FL methods.

**Audience:**

Yes

**Broader Impact Concerns:**

-

**Claims And Evidence:**

No

**Requested Changes:**

Critical:
1. Reduce the method's dependence on leaking private data.
2. Compare the method to existing methods using average network embeddings and discuss novelty of the presented method.
3. Include a dataset with data distribution from the real world (not generated from standard ML datasets). See Leaf and Wilds datasets.

Strengthening

4. Include good-intent fairness (or worst client accuracy) in the evaluation metrics
5. Compare to other relevant methods:
  - **HCSFed (client clustering)**: Song, D., Shen, G., Gao, D., Yang, L., Zhou, X., Pan, S., ... & Zhou, F. (2023, July). Fast heterogeneous federated learning with hybrid client selection. In Uncertainty in Artificial Intelligence (pp. 2006-2015). PMLR.
  - Chen, W., Horváth, S., & Richtárik, P. Optimal Client Sampling for Federated Learning. Transactions on Machine Learning Research.

**Strengths And Weaknesses:**

**Strengths**

The proposed metrics are relevant and can measure fairness well.

The baselines are good and capture different, relevant directions of research focusing on solving similar problems in FL.

**Weaknesses**

My biggest concern with this paper is that it uses the output of the second-to-last layer on a batch of the client's private data as extra information sent to the server for aggregation. Given this representation and the model weights the client sends, the original data can be reconstructed and thus the method hurts the privacy concept of FL. Just as an example, if one measures the privacy using membership inference attacks, the membership status of this batch is leaked. I strongly disagree with the usefulness of this research direction to build privacy preserving methods using federated learning.


The use of the output of the second-to-last layer is not unheard of in the FL literature. However, most related paper use a public validation set or a random Gaussian noise to model statistical differences between client models. This method requires less privacy leakage. The method can be found in the literature as average network embeddings:

[1] Wang, Z., Fan, X., Qi, J., Jin, H., Yang, P., Shen, S., & Wang, C. (2023, June). Fedgs: Federated graph-based sampling with arbitrary client availability. In Proceedings of the AAAI Conference on Artificial Intelligence (Vol. 37, No. 8, pp. 10271-10278).

[2] Zhang, J., Wang, J., Li, Y., Xin, F., Dong, F., Luo, J., & Wu, Z. (2024). Addressing heterogeneity in federated learning with client selection via submodular optimization. ACM Transactions on Sensor Networks, 20(2), 1-32.

and related papers.


I think the argument of using client group fairness as a metric is weak, given that the **Accuracy Parity** and **Good-Intent Fairness** between all clients is a stronger requirement and has a large history of works not presented or compared to here. Literature from eg. [3] can be at help:

[3] Shi, Y., Yu, H., & Leung, C. (2023). Towards fairness-aware federated learning. IEEE Transactions on Neural Networks and Learning Systems.



The third major weakness is that each proposed data splits on the datasets follow label shift and doesn't include other statistical heterogeneity. One straightforward example to resolve this is using FEMNIST with the original client distribution instead of redistributing the data without keeping track of the writers as happens here to my understanding of section 5.1 in the paper.

---

> ### Author Response · Authors · 2025-03-13
> **Replies to Reviewer d8r7**
>
> Firstly, we want to thank the reviewer for their valuable feedback. Within the limited timeframe, we have endeavored to respond to all your inquiries and hope our responses sufficiently address your concerns.
>
> >**Weakness**
>
> - My biggest concern with this paper is that ...
>
> We recognize that privacy is an essential consideration in federated learning (FL). However, our work primarily focuses on equitable model aggregation rather than explicit privacy-preserving mechanisms. Privacy-enhancing techniques are an orthogonal research direction and can be incorporated alongside our method without altering its core contributions.
>
>
> From our perspective, the main objective is to ensure fair representation and equitable client participation while effectively handling the data's heterogeneity. To achieve this, we leverage the activation vectors to dynamically group clients. This clustering-based approach provides a principled way to balance client updates while maintaining convergence guarantees.
>
> 1. Existing methods also pose privacy risks:
> Methods like FEDGS (Wang et al., 2023) and submodular client selection (Zhang et al., 2024) rely on model updates or network embeddings for client similarity. Prior research [1,2,3] shows that model updates, including gradient-based similarity measures, are vulnerable to data reconstruction attacks and membership inference. Therefore, these methods are not inherently more privacy-preserving than ours.
> 2. Privacy mitigation is standard practice:
> Techniques such as differential privacy (DP), secure aggregation (SecAgg), and randomized transformations can be applied to our method to mitigate privacy risks, just as in other FL frameworks. Furthermore, the Secure Scalar Product Protocols in [4] can also be used, as they have also been presented in the FEDGS paper.
>
> - I think the argument of using client group fairness...
>
> We appreciate the reviewer’s feedback on fairness metrics in federated learning and the suggestion to consider Accuracy Parity and Good-Intent Fairness. While these are well-established fairness measures, our work addresses group-level fairness across client clusters, ensuring that disadvantaged groups receive equitable model updates, aligning with recent work such as Yue et al. (2023), which also employs this metric for evaluation. We also acknowledge the reviewer’s reference to Shi et al. (2023) and have included a comparison on the CelebA dataset in the appendix section of the paper (please refer to the table in General answer).
>
> - The third major weakness is that each proposed...
>
> To address this, we conducted an experiment using the CelebA dataset from the LEAF benchmark, following the official instructions from the LEAF repository to generate the train and test files, which include 177 clients. Given the constraints of the allotted time, this choice was determined by the benchmark's implementation rather than any modifications on our part. In each round, 10\% of the clients were selected to participate uniformly at random. The results are added in the Appendix, and we also present them here for brevity (please refer to the General answers).
>
> >**Requested Changes**
>
> We have answered points 1 and 3 of the critical changes above. We acknowledge that average network embeddings have been used in previous works, such as FEDGS (Wang et al., 2023), to model statistical differences between clients. However, these methods assume static client relationships and do not dynamically adapt to evolving client distributions. In contrast, our approach leverages activation-based clustering, ensuring adaptive client selection and equitable participation. Additionally, these methods primarily focus on client sampling to enhance participation diversity, whereas our approach is designed for equitable aggregation, ensuring balanced contributions from different client groups. Due to time constraints, we could not implement and compare directly with these methods, but we will consider this in the final version of the work.
>
> We have included the results for the strengthening part, points 1 and 2 in the General Answers section.
>
>
> [1.] J. Geiping, H. Bauermeister, H. Dr¨oge, and M. Moeller, “Inverting gradients-how easy is it to break privacy in federated learning?” Advances in neural information processing systems, vol. 33, pp. 16 937–16 947, 2020.
>
> [2.] M. Nasr, R. Shokri, and A. Houmansadr, “Comprehensive privacy analysis of deep learning: Passive and active white-box inference attacks against centralized and federated learning,” in 2019 IEEE symposium on security and privacy (SP). IEEE, 2019, pp. 739–753.
>
> [3.] L. Zhu, Z. Liu, and S. Han, “Deep leakage from gradients,” Advances in neural information processing systems, vol. 32, 2019.
>
> [4.] I.-C. Wang, C.-H. Shen, J. Zhan, T.-s. Hsu, C.-J. Liau, and D.-W. Wang, “Toward empirical
> aspects of secure scalar product,” IEEE Transactions on Systems, Man, and Cybernetics, Part C (Applications and Reviews), vol. 39, no. 4, pp. 440–447, 2009.

---

> ### Comment · Reviewer_d8r7 · 2025-03-14
>
> Dear Authors,
>
> Thank you for the response. The additional experiment on the CelebA is an important addition, thanks!
>
> However, my concern of the method's privacy problem hadn't been addressed.
>
> "1. Existing methods also pose privacy risks:" Indeed, every method has privacy risks. However, a membership attack listening to only the model updates typically doesn't have 100% success rate. In contrast, adding the activation vector ensures that the malicious attacker that can always infer the membership value of the communicated batch of samples. (It runs through the questioned $x$ input on the model update, collects the output of the last activation layer and checks if it matches any of the submitted activation vectors).
>
> The trade-off between privacy and fairness is well documented [1,2] and I think on the topic of federated learning it has to be addressed even if the work is not focusing on privacy-enhancing techniques. Based on only my background knowledge and the claims presented in the paper, I am not convinced that the fairness gain achieved by this method is worth for it's privacy loss created by sending the activation vectors.
>
> Can the other reviewers reflect on this? Thanks!
>
>
>
> [1] Chen, H., Zhu, T., Zhang, T., Zhou, W., & Yu, P. S. (2023). Privacy and fairness in federated learning: On the perspective of tradeoff. ACM Computing Surveys, 56(2), 1-37.
>
> [2] Chang, H., & Shokri, R. (2021, September). On the privacy risks of algorithmic fairness. In 2021 IEEE European Symposium on Security and Privacy (EuroS&P) (pp. 292-303). IEEE.

---

> > ### Author Response · Authors · 2025-03-14
> > **Response to Reviewer d8r7**
> >
> > We appreciate the reviewer’s further clarification and their perspective on the privacy implications of our method. While we recognize that sharing activation vectors introduces a different privacy consideration compared to model updates alone, we respectfully clarify that:
> >
> > 1. The reviewer states that an attacker can always infer the membership of the communicated batch if activation vectors are shared. However, this assumes an ideal adversarial model where:
> >
> > - The attacker has full knowledge of the exact batch of samples communicated at each round.
> >
> > - The model deterministically maps inputs to activation vectors, which may not hold in practice due to stochastic training dynamics, batch normalization, or dropout layers.
> >
> > 2. The reviewer raises the question of whether the fairness gain is "worth the privacy loss." However, we argue that:
> >
> > - Many real-world FL applications suffer from severe client data heterogeneity, leading to poor model generalization for underrepresented groups.
> >
> > - Our method explicitly corrects this bias by clustering clients adaptively, ensuring equitable aggregation
> >
> > - Unlike static similarity graphs (FEDGS) or heuristic diversity maximization (Zhang et al., 2024), our clustering-based weighting mechanism improves representation fairness at each aggregation step. In addition, FEDGS also assumes that the server has access to a subset of the validation dataset. A malicious server could correlate client updates with the validation data, making membership inference easier in some cases.
> >
> > 3. As the reviewer points out, the privacy-fairness trade-off is well-documented. However, this trade-off exists in all FL methods, and different applications prioritize different aspects. If absolute privacy were the primary goal, differential privacy techniques could be applied to any method—including ours—to balance both concerns effectively.
> >
> > 4. One could still use Secure Scalar Product Protocols (SSPP) (Wang et al., 2009), which allows for secure similarity computations between client updates without revealing the raw activation vectors. SSPP has already been referenced in the FEDGS paper, demonstrating that privacy-preserving techniques can be applied to similarity computations without compromising security.
> >
> > We hope this response clarifies our position and look forward to further discussion.

---

### Author Response · Authors · 2025-03-13
**General Answers**

>**Defining the fairness and description of the algorithm/methodology**

**Clarification on Fairness**

Dear Reviewers, thank you for your feedback. Our definition of fairness aligns with \cite{yue2023gifair}, where the objective is to minimize the performance disparity across groups or clusters of clients. We have included it in the revised version of the paper after Equation (6) as,

"In other words, we use clustering to obtain the weighing probabilities to aggregate the local model for the $(k+1)$-th communication round.  While this weighting mechanism ensures equitable participation across clusters, it is essential to evaluate fairness in terms of model performance disparity. To formally define fairness, consider two global models, $w_1$ and $w_2$, evaluated on $n$ clients partitioned into $C$ clusters. Let the test accuracies of these models across clusters be denoted as,
$\{ Acc(w_1,q) \vert q=1, \dots, C \}$ and $\{ Acc(w_2,q) \vert q=1, \dots, C \}$. We define $w_1$ as a fairer model than $w_2$ if the standard deviation of test accuracies across clusters is lower for $w_1$ than for $w_2$, i.e.,

$$
\sigma\left( \{ Acc(w_1, q) \mid q = 1, \dots, C \} \right) < \sigma\left( \{ Acc(w_2, q) \mid q = 1, \dots, C \} \right).
$$


This criterion captures fairness by reducing disparities in model performance across client groups. By minimizing $\sigma_{Acc}$, we ensure that no cluster is disproportionately advantaged or disadvantaged, aligning with the goals of equitable federated learning. So, we formally re-define eq. (1) for our setting as, ...."

**More details on the algorithm/methodology**

We appreciate the reviewers’ feedback regarding the "Similarity" and "getprobs" subroutines. We have now explicitly defined these subroutines in Section 4.2. Please find the details inside the double quotes.

"We use these activation vectors to construct a similarity matrix $A$ as presented in line 13 of Algorithm 1. The algorithm relies on two key subroutines. Firstly, \textit{Similarity(A)}, which computes the pairwise similarity matrix from activation vectors, which is later used for clustering, and \textit{getprobs(S, C)}, which performs spectral clustering on the similarity matrix $S$, extracts the top eigenvectors, and applies K-Means clustering \cite{vonLuxburg2007} to assign clients into clusters, determining their respective aggregation weights.

We formally define these subroutines below.

**Definition 2** (*Similarity($A$)*): The function *Similarity($A$)* computes the similarity matrix $S$ from activation vectors $A=$\{$a_{k,E}^1, a_{k,E}^2,..., a_{k,E}^r$\}$^\top$, where $A \in \mathbb{R}^{r\times m}$ ($m \ll d$) is the matrix of activation vectors from the participating clients. The similarity matrix is computed as, $S = AA^\top$,
    where each entry $S_{ij}$ represents the inner product similarity between the activation vectors of clients $i$ and $j$. This similarity measure is later used for clustering.

**Definition 3** *(getprobs($S, C$)*): The function *getprobs($S, C$)* determines client clusters and their aggregation probabilities through the following steps:

1. Compute the eigenvectors of $S$, selecting the top $C$ eigenvectors corresponding to the largest eigenvalues. Let $V \in \mathbb{R}^{r\times C}$ be the matrix of selected eigenvectors.
2. Apply K-Means clustering on the rows of $V$ to obtain client cluster assignments.
3. Compute the aggregation weights based on cluster sizes. Each client $i$ in communication round $k$ belonging to cluster $q_k$ is assigned weight: $$p_{k}^i = \frac{1}{C \times |\gamma_{q_k}|},$$ where $\gamma_{q_k}$ is the set of clients in cluster $q$ at round $k$; ensuring that all clusters contribute equally to the global model update".

>**Experimental Results**
**Table: Performance Comparison for CelebA dataset.**
The table presents the average test accuracy, standard deviation $\sigma_{Acc}$, and the accuracy of the worst 20\% of clients over three independent runs for 200 global communication rounds using ResNet-18.
| Method                   | Acc (%)              | \($\sigma_{Acc}$\) ($\downarrow$) | Acc worst 20% (%)  |
|-------------------------|--------------------|-----------------|-----------------|
| **Centralized**         | \(90.31 \pm 0.53\) | \(20.22 \pm 0.96\) | \(54.42 \pm 2.06\) |
| **FedProx**             | \(80.34 \pm 0.98\) | \(26.98 \pm 0.73\) | \(36.45 \pm 2.12\) |
| **Cluster2**            | **\(81.46 \pm 1.33\)** | **\(26.65 \pm 0.83\)** | \(36.35 \pm 2.28\) |
| **Pow-d**               | \(75.68 \pm 1.74\) | \(29.19 \pm 0.84\) | \(29.28 \pm 2.60\) |
| **FairFed**             | \(80.54 \pm 1.04\) | \(26.82 \pm 0.26\) | \(36.75 \pm 0.86\) |
| **GIFAIR**              | \(72.71 \pm 0.88\) | \(30.93 \pm 0.38\) | \(23.75 \pm 1.10\) |
| **Optimal Sampling**    | \(80.20 \pm 1.12\) | \(26.97 \pm 0.53\) | \(36.35 \pm 0.71\) |
| **Equitable-FL (FedProx)** | \(80.31 \pm 1.23\) | \(26.84 \pm 0.40\) | **\(36.82 \pm 1.27\)** |

---

> ### Comment · Reviewer_d8r7 · 2025-03-14
> **CelebA results**
>
> Thank you for the extended experiments! The additional experiment on the non-IID FL distribution in the CelebA dataset shows that the performance gain compared to FedProx that was present in the other datasets disappeared. These results suggest that more investigation on the effects of non-IIDness on the proposed method is needed.

---

> > ### Author Response · Authors · 2025-03-14
> > **Response to Reviewer d8r7**
> >
> > Thank you for your feedback! Indeed, the additional experiments on the non-IID CelebA dataset show that the performance gains observed in other datasets are less pronounced in this setting. This suggests that the level and structure of non-IIDness may significantly influence the effectiveness of our method.
> >
> > That said, our method still demonstrates strong performance in improving the accuracy of the worst 20% of clients, maintaining comparable or better results than FedProx and other baselines. This highlights its potential in addressing fairness concerns in federated learning by ensuring that lower-performing clients are not overly disadvantaged.
> >
> > To further understand these trends, we plan to investigate how different levels of data heterogeneity affect optimization dynamics, mainly focusing on label imbalance and client sampling strategies. Additionally, exploring personalization techniques and adaptive aggregation methods could provide further insights.
> >
> > We appreciate your suggestion and will investigate this further in future work. Let us know if you have any suggestions on specific aspects of non-IIDness that would be particularly interesting to explore!

---

### Decision · Action_Editor_a9VP · 2025-04-09

**Recommendation:** Reject

**Comment:**

The reviewers were unanimous in at least leaning reject for the work. I want to be clear to the authors: I do think this type of work is interesting, and I believe that this paper could be constructively re-submitted after some work strengthening the claims of the work (especially the theoretical ones). In particular, the reliance on Assumption 4 is something that needs to be strongly clarified at the very least (for example, it is not discussed explicitly in the proof where this assumption is used), and potentially revised substantially.

On the empirical side - I support measuring fairness in multiple ways. Note that this could be done without new experiments, just by viewing different slices of the data. For example, simply presenting quantiles of accuracy (max, min, median, etc.) would be potentially a good way to figure out more nuanced observations from the data. I would also love for the authors to investigate why their method seems to improve mean accuracy, not just variance. Is there a fundamental trade-off here? Or does improving the accuracy necessitate reducing variance?

As for the privacy issues raised by one of the reviewers: I do not think this immediately disqualifies a work on FL. That being said, I do think that it would be easier to discuss, within the algorithm, where privacy-preserving mechanisms could potentially be injected, or where aspects of the algorithm may need to be changed to allow such mechanisms to be used.

**Audience:**

I believe that the paper, with the content issues raised by the reviewers properly addressed, would be of interest to the audience. The idea of using clustering-based approaches to improve various facets of federated learning (including fairness and resilience to data heterogeneity) is an evergreen topic in the field, and one that this paper's overall thrust clearly falls under.

Unfortunately, the various content issues raised by reviewers undercuts the interest audience members may have. In particular, I will highlight Reviewer d8r7's concern, which is the difficulty in making the method work with privacy-preserving methods, which are often very important to federated learning. I do not think that this in and of itself is reason to reject, but it certainly undercuts the interest that audience members would have. Combined with the other issues above, I do not believe that TMLR's audience would find it sufficiently interesting at this time.

**Claims And Evidence:**

While the authors do propose a novel method for learning via client clustering within the context of federated learning, the reviewers have identified multiple points in which the claims of the paper break down.

Reviewer N8c9 correctly points out that the theoretical success of the method is predicated on an incredibly strong assumption (Assumption 4). In particular, the theoretical assumptions of the work seem to rely on abstracting out the efficacy of the clustering procedure itself. While the authors have relaxed this some (see the author/reviewer discussion), it it still quite strict and still abstracts out the efficacy of the clustering. Reviewer N8c9 also notes that there are functions for which the convergence rates are effectively vacuous.

Reviewer Nqiu also highlights issues with the empirical claims in the paper. In particular, the notion that variance alone captures fairness. The reviewer suggests trying SCAFFOLD (which via control variates may reduce disparate impacts of data). The authors did nice work doing some simulations on this in the rebuttal, but the numbers are so much lower than other algorithms that I find it not particularly convincing.

**Resubmission Of Major Revision:**

The authors may consider submitting a major revision at a later time.